# A novel synaptic plasticity rule explains homeostasis of neuromuscular transmission

Gilles Ouanounou*, Gérard Baux, Thierry Bal

Unité de Neuroscience Information et Complexité, Centre National de la Recherche Scientifique, FRE 3693, Gif-sur-Yvette, France

**Abstract** Excitability differs among muscle fibers and undergoes continuous changes during development and growth, yet the neuromuscular synapse maintains a remarkable fidelity of execution. Here we show in two evolutionarily distant vertebrates (*Xenopus laevis* cell culture and mouse nerve-muscle ex-vivo) that the skeletal muscle cell constantly senses, through two identified calcium signals, synaptic events and their efficacy in eliciting spikes. These sensors trigger retrograde signal(s) that control presynaptic neurotransmitter release, resulting in synaptic potentiation or depression. In the absence of spikes, synaptic events trigger potentiation. Once the synapse is sufficiently strong to initiate spiking, the occurrence of these spikes activates a negative retrograde feedback. These opposing signals dynamically balance the synapse in order to continuously adjust neurotransmitter release to a level matching current muscle cell excitability.

## Introduction

In the nervous system, presynaptic neurotransmitter release, postsynaptic receptors, and postsynaptic excitability can be modulated to bi-directionally and durably change synaptic efficacy. There is a large diversity of plasticity processes, which together shape the neuronal network (*Changeux and Danchin, 1976*; *Goda and Davis, 2003*; *Munz et al., 2014*) and tune its properties (*Nelson and Turrigiano, 2008*). In the brain, concomitantly to various forms of associative plasticity which sustain memory and learning, homeostatic plasticity processes are proposed to restrain the mean level of neuronal activity within a physiological regime, and to maintain the stability of recurrent network activity that can be challenged by associative plasticity (*Turrigiano and Nelson, 2004*; *Macleod and Zinsmaier, 2006*; *Marder and Goaillard, 2006*; *Turrigiano, 2007*; *Nelson and Turrigiano, 2008*). Homeostatic plasticity has been extensively studied at the neuromuscular synapse, in particular at the glutamatergic neuromuscular junction (NMJ) in *Drosophila* (*Frank, 2014*), due to the robustness of the homeostatic control of synaptic transmission and the great accessibility of the experimental model to genetic manipulations.

In vertebrates, each skeletal muscle fiber is mono-innervated (*Sanes and Lichtman, 1999*), and each single presynaptic action potential (AP) induces one postsynaptic AP, corresponding to a unity synaptic gain (ratio between the numbers of post- and presynaptic APs equal to 1) (*Wood and Slater, 2001*). The stability of the gain implies that presynaptic neurotransmitter release and/or postsynaptic receptors adapt the effective synaptic strength to the excitability of the muscle fiber, which depends on the fiber characteristics, and presumably decreases with growth and exercise (*Turrigiano, 2007*). In adult vertebrate skeletal muscles, cholinergic nicotinic receptors are clustered in the synaptic region. Expression and location of nicotinic receptors have been shown to depend not only on agrin (*McMahan, 1990*; *Hall and Sanes, 1993*; *Gautam et al., 1995*; *1996*; *Sandrock et al., 1997*) but also on activity (*Lømo, 2003*), suggesting their possible role as adjustment variables in

*For correspondence: gilles.
ouanounou@unic.cnrs-gif.fr

**Competing interests:** The authors declare that no competing interests exist.

**eLife digest** Nerve cells communicate with each other, and with targets such as muscle cells, at junctions called synapses. The nerve cell before the synapses releases a chemical called a neurotransmitter, which binds to receptors on the cell after the synapses. However, the first cell cannot determine by itself whether it is releasing the correct amount of neurotransmitter to activate its partner. For this, it requires feedback from the second cell.

This feedback is particularly important at synapses between nerve cells and muscle cells, which are known as neuromuscular junctions. The likelihood that a given amount of transmitter will activate a muscle cell can vary with age and after exercise. Muscle cells must therefore be able to instruct their nerve cell partners to increase or decrease neurotransmitter release to accommodate these changes.

Ouanounou et al. have now identified the mechanism by which muscle cells determine whether nerve cells are releasing an appropriate amount of neurotransmitter. Experiments in two distantly related animals – mice and embryos from a frog called *Xenopus* – revealed that muscle cells use two calcium-based signals. The first is the flow of calcium ions into the muscle cell in response to binding of neurotransmitter to receptors at the synapses: this tells the muscle cell how active the nerve cell is. The second is the release of calcium ions from internal stores inside the muscle cell: this occurs whenever neurotransmitter release is sufficient to activate the muscle cell.

In response to the first calcium signal, the muscle cell sends positive feedback to the neuron, telling it to increase neurotransmitter release further. In response to the second signal, the muscle cell sends negative feedback to reduce neurotransmitter release. Thus, when neurotransmitter release is not enough to activate the muscle, positive feedback dominates and neurotransmitter release increases. However, when the muscle is activated, the two types of feedback act in balance to maintain efficient communication across the synapse.

The next steps are to identify the cell signaling cascades that are mobilized by the two calcium signals, including the specific molecule (or molecules) that regulate neurotransmitter release.

---

the control of synaptic strength. In the recent years however, thorough studies of the glutamatergic NMJ in *Drosophila* revealed that presynaptic regulation of neurotransmitter release is certainly a major adjustment variable for synaptic homeostasis (*Davis and Müller, 2015*). In such 'presynaptic homeostasis', increase in neurotransmitter release counterbalances a genetically-induced decrease of the postsynaptic sensitivity to glutamate (*Petersen et al., 1997*; *Davis et al., 1998*; *DiAntonio et al., 1999*), a genetically-induced increase of postsynaptic input conductance (*Paradis et al., 2001*) or a pharmacological blockade of postsynaptic receptors (*Frank et al., 2006*), thus resulting in the strict maintenance of the level of evoked postsynaptic depolarization. Presynaptic compensation of postsynaptic excitability changes implies the existence of a retrograde feedback process from the innervated muscle fiber to the motor neuron. Particular attention was paid to the determination of the molecular targets of the homeostatic retrograde signaling and of the presynaptic cell signaling involved. It appeared that both the readily releasable vesicle pool (*Müller et al., 2012*; *2015*) and the presynaptic voltage-gated $Ca^{2+}$ channels (*Müller and Davis, 2012*) are targeted to promote glutamate release when postsynaptic excitability is reduced. The abundance of voltage-gated $Ca^{2+}$ channels can also be decreased in response to a vesicular content increase (*Gaviño et al., 2015*), confirming the bi-directionality of the process. In *Drosophila*, endostatin is a candidate trans-synaptic factor for the retrograde feedback targeting presynaptic neurotransmitter release (*Wang et al., 2014*). Upstream from the retrograde factors, the postsynaptic kinases, CAMKII (*Haghighi et al., 2003*) and TOR (*Penney et al., 2012*), pathways are necessary for a functional homeostatic control at the *Drosophila* neuromuscular transmission.

An abundant literature is thus progressively assembling pieces of the signaling puzzle underlying the interactions between the motor neuron and the muscle cell for the control of synaptic strength. However, besides the nature of the retrograde feedback and its presynaptic targets, a mechanism for the evaluation of the synaptic strength should be present at the postsynaptic level. The nature and dynamics of these mechanisms, primarily sensing synaptic strength in the muscle cell and

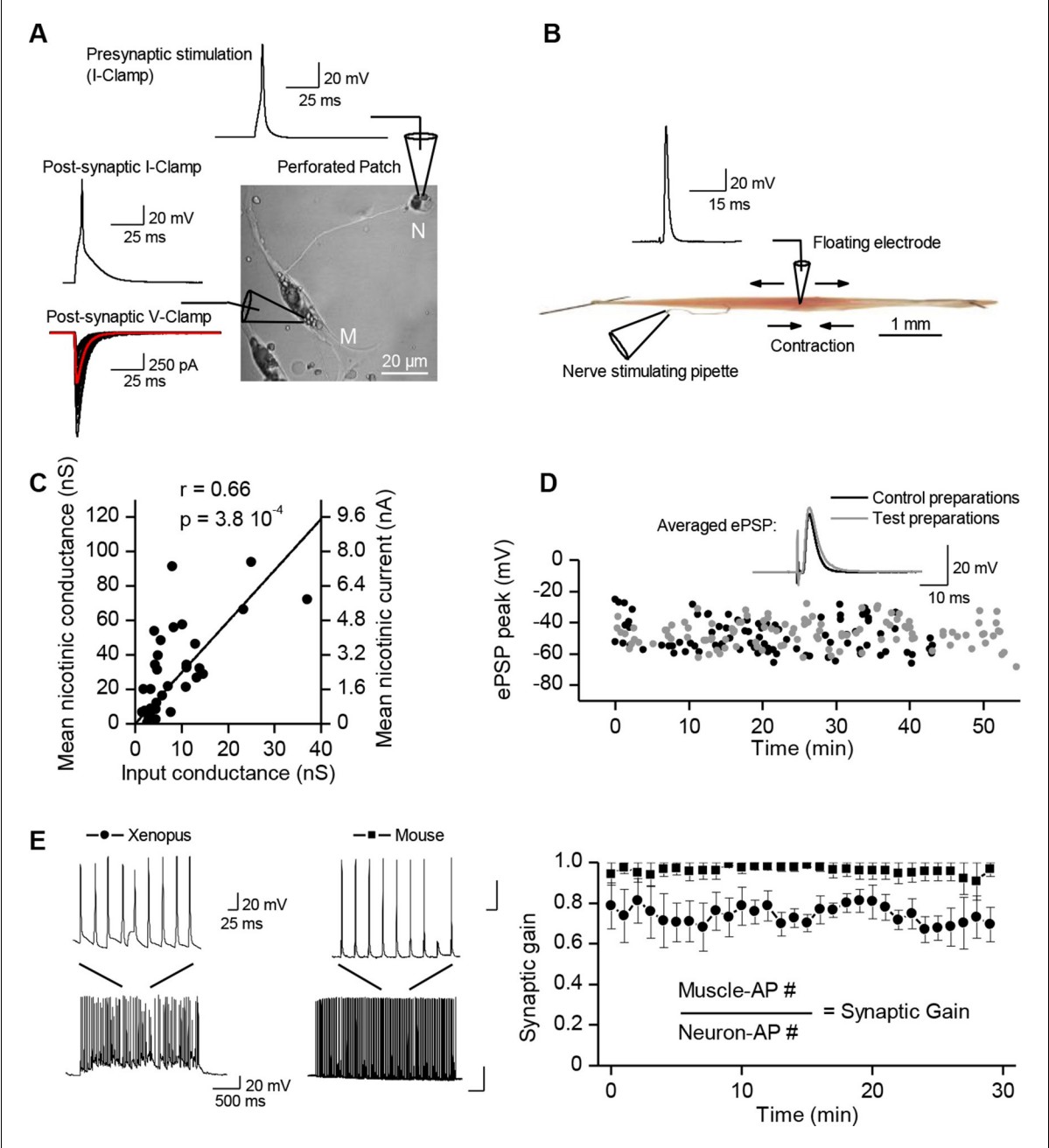

**Figure 1.** Synaptic transmission is homeostatically regulated. (**A**) Perforated patch-clamp on *Xenopus* neuron (N) and muscle cell (M) in primary culture. Presynaptic APs were triggered with current steps. Postsynaptic APs were recorded under current-clamp and nicotinic synaptic currents under voltage-clamp (-80 mV). (**B**) Intracellular recording in soleus muscle fibers from an adult mouse using a floating sharp electrode (see Materials and Methods and *Figure 1—figure supplement 1*). (**C**) Nicotinic conductance calculated from averaged ePSCs (n = 30 ePSCs for each dot) in different *Xenopus* muscle cells as a function of their input conductance. The black line shows the linear regression. (**D**) In mice, membrane potential reached by the ePSP in individual FDB muscle fibers after treatment with μ-conotoxin GIIIB, in absence of burst stimulation of the nerve (black dots, n= 88 fibers, 2 muscles, 2 mice), and in test preparations (grey dots, n = 108 fibers, 2 muscles, 2 mice) for which the nerve was burst stimulated prior to conotoxin treatment (15 bursts in 10 min, each of 120 events at 30 Hz). (**E**) Mean synaptic gain at *Xenopus* synapses (dots, n = 5 synaptic connections) and in mouse neuromuscular junctions (squares, n = 4 muscle fibers from different mice) during chronic bursts of presynaptic stimulation (bursts of 80 to 120 pulses, 30 Hz for 30 min).

The following figure supplements are available for figure 1:

**Figure supplement 1.** Floating electrode.

*Figure 1 continued on next page*

*Figure 1 continued*

**Figure supplement 2.** Mouse muscles fibers have a wide range of input conductances.

**Figure supplement 3.** Characterization of the K+ conductances that determine the *Xenopus* muscle cell input conductance.

initiating retrograde feedback, are key issues of homeostatic plasticity that still remain enigmatic in both invertebrates and vertebrates.

Here we investigate this issue in vertebrate (*Xenopus* and mouse) cholinergic transmission. Motor neurons and muscle cells of *Xenopus laevis* embryos establish functional neuromuscular synapses (*Tabti et al., 1998*) in primary co-culture. Standard intracellular recording from muscle cells is possible in *Xenopus* culture owing to the small size of the cells (*Figure 1A*, perforated patch-clamp), while it is not possible in muscle cells from adult mice, due to macroscopic movements during contraction. To circumvent this issue and maintain intracellular recordings in moving tissue, we adapted a 'floating electrode' device from previous methods invented for muscles (*Woodbury and Brady, 1956*) or neurons (*Kunze, 1998*). We applied this method to mouse ex vivo nerve-skeletal muscle preparations (*Figure 1B*, Methods, and *Figure 1—figure supplement 1*). This approach preserved the physiological conditions, necessary to reveal the control exercised by the muscle cell over the neuromuscular synaptic transmission, a role that might have been previously overlooked in vertebrates due to commonly used high concentrations of the nicotinic receptor antagonist curare to prevent contraction.

## Results

### A ubiquitous unity synaptic gain

Synaptic efficacy is linked to the ability of evoked postsynaptic potentials (ePSP) to reach firing threshold. In vertebrate muscle cells, the amplitude of PSPs depends on the ratio between the nicotinic conductance activated by acetylcholine (ACh) and the muscle input conductance, which is largely due to resting $K^+$ leak currents. Fibers in flexor digitorum brevis (FDB), extensor digitorum longus and soleus mouse muscles have a wide distribution of input conductances (*Figure 1—figure supplement 2*). A similar distribution in *Xenopus* muscle cells (*Figure 1C*, X axis) is linked to the diversity of membrane surfaces with homogenous conductance density ($196 \pm 14$ pS/pF, n=19, *Figure 1—figure supplement 3*). Despite these differences in muscle cell excitability, a single presynaptic AP triggered a single postsynaptic AP, and a contraction, in all tested cells from *Xenopus* and mouse (*Figure 1*). In *Xenopus* cultures, we found that the nicotinic conductance activated in muscle cells by spike-evoked ACh release varied linearly with the postsynaptic input conductances measured in a population of synaptic neuron/muscle cell pairs (*Figure 1C*), and resulted in comparable ePSP (evoked PSP) voltage amplitudes. Similarly, in mouse muscles, we assessed ePSP characteristics after exogenous application of μ-conotoxin GIIIB which selectively blocks $Na^+_v1.4$ channels involved in the muscle cell AP (*Cruz et al., 1985*; *Li et al., 2003*). This manipulation does not affect nerve APs, since this $Na^+$ channel subtype is not critical for spiking in motor neurons. Single-spike-evoked PSPs exhibited a narrow range of voltage magnitudes (*Figure 1D*, in FDB muscle). The comparable amplitude of ePSPs, despite widely ranging resting conductances in the postsynaptic muscle cells, indicates that the neuromuscular synapse exhibits robust plasticity that regulates its functional strength within a narrow range.

### A stable synaptic gain

How does the neuromuscular synapse adjust its strength depending upon the excitability of the postsynaptic muscle fiber? Previous research reported that in *Xenopus* cell cultures, burst activation of presynaptic input results in marked enhancement of ACh release, in both voltage-clamp and current-clamp conditions (*Wan and Poo, 1999*). We reproduced this result in the specific condition of post-synaptic voltage-clamp (not shown) but not in that of current-clamp where we found instead absence of potentiation. To observe such absence in current-clamp, we believe that it is essential

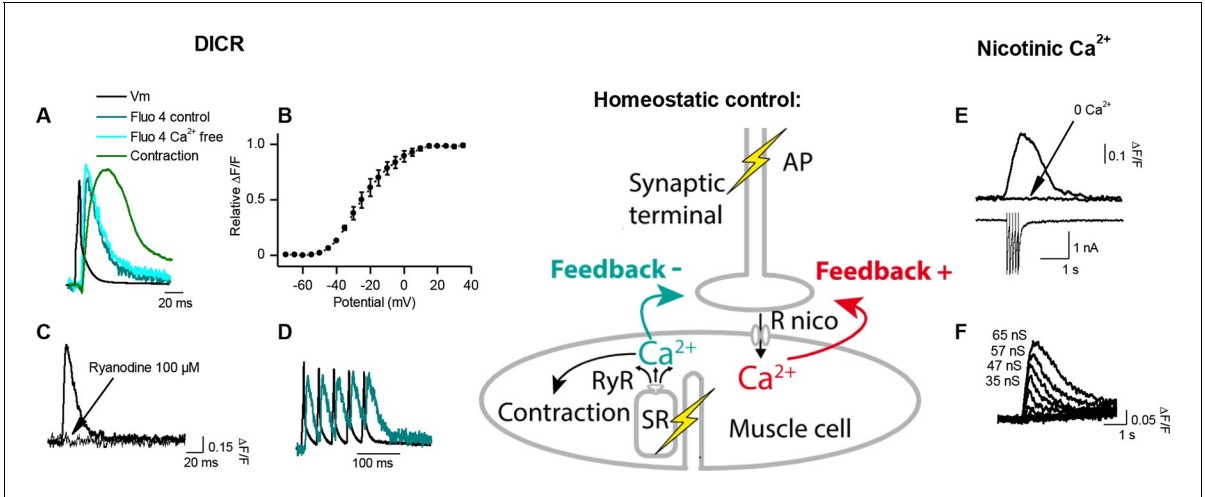

**Figure 2.** Calcium signaling in *Xenopus* muscle cell and synaptic homeostasis. The middle scheme depicts our model of homeostatic control of the synaptic strength. The nicotinic $Ca^{2+}$ influx elicits a positive retrograde feedback signal (red arrow) on the presynaptic neurotransmitter release. The positive feedback is balanced by a negative retrograde feedback signal (blue arrow) triggered by the muscle AP-induced DICR. (DICR = depolarization-induced calcium release, AP = action potential, RyR = ryanodine receptor, SR = sarcoplasmic reticulum, R nico = nicotinic receptors). (**A**) Independence of the DICR signal from external $Ca^{2+}$. AP (black trace, induced with a brief postsynaptic current step), $Ca^{2+}$ dye (Fluo4) relative fluorescence in standard external medium (dark blue trace) and in $Ca^{2+}$ free medium (light blue trace), and qualitative measure of the cell contraction (green trace, see Materials and methods). (**B**) Voltage-dependence of the DICR signal. The Fluo4 fluorescence was measured at the steady-state of the $Ca^{2+}$ signal during a classical voltage-step protocol from a holding potential of -80 mV and averaged (n = 3 cells). (**C**) Blockade of the DICR signal by loading the muscle cell with 100 μM ryanodine. (**D**) Representative example of the DICR signal (blue trace) during repetitive muscle APs (black trace). (**E**) $Ca^{2+}$ build-up upon nicotinic receptor activation. Fluo4 signal (upper traces) in control and in $Ca^{2+}$ free medium, during iontophoretic ACh applications (5 pulses) under postsynaptic voltage-clamp (-80 mV). The lower trace shows the nicotinic currents. (**F**) Dependence of the nicotinic $Ca^{2+}$ build-up on the nicotinic conductance with increasing iontophoretic ACh applications at a constant membrane potential (holding potential –80 mV).

The following figure supplement is available for figure 2:

**Figure supplement 1.** Ionic permeability of the *Xenopus* nicotinic receptor-channel.

that all or most of the ePSPs evoked by the conditioning burst, reach threshold for spike initiation in the postsynaptic cell. This was not the case in the only example of current-clamp presented by *Wan and Poo 1999* (see their *Figure 1A*(i)). Homeostatic plasticity rules suggest in fact that this form of plasticity should not occur in the presence of synaptic transmission that is effective in generating a muscle AP. To test this hypothesis, we initiated burst stimulation of motor neurons in both *Xenopus* and mice –during current-clamp recordings– in order to allow voltage responses and APs in the muscle cells. In these physiological conditions and despite the chronic burst activity, no significant change in the synaptic gain was visible. The probability for a synaptic event to induce a postsynaptic AP was close to unity and kept constant (*Figure 1E*). In *Xenopus* however, test evoked postsynaptic currents (ePSCs) were recorded under brief periods of postsynaptic voltage-clamp before and after the 30 min of chronic activity under current-clamp, and showed a slight relative decrease of 0.2 in the averaged ePSCs amplitude (see first bar in *Figure 3C*). In mice, the average ePSPs were not changed by high-frequency conditioning stimulations (*Figure 1D* and *3F*). Altogether, these observations suggest that synapses able to trigger postsynaptic APs do not exhibit strong plasticity, and that the occurrence of muscle APs may be involved in the homeostatic mechanisms adjusting and maintaining the synaptic strength in concordance with the postsynaptic input conductance.

## Activity-dependent $Ca^{2+}$ signaling and synaptic homeostasis

How might the strength of synaptic transmission depend upon the occurrence of an AP in the postsynaptic muscle cell? We reasoned that there may be two distinct postsynaptic $Ca^{2+}$ signaling pathways involved in this homeostatic regulation: a postsynaptic reporter of the presynaptic activity that

signals the occurrence of a synaptic event, and a postsynaptic reporter that signals the occurrence of a postsynaptic AP. The postsynaptic $Ca^{2+}$ build up (*Figure 2E–F*) due to a high $Ca^{2+}$ permeability of the nicotinic receptor-channel (*Fucile et al., 2006*) ($PCa^{2+}/PNa^{+}$=0.23 in *Xenopus*, *Figure 2—figure supplement 1*) is a good candidate for signaling the occurrence of synaptic events. Furthermore, in vertebrates, the skeletal muscle cells exhibit a specific $Ca^{2+}$ signal linked to the AP and involved in the excitation-contraction coupling, *i.e.* 'the depolarization-induced $Ca^{2+}$ release' (DICR). The DICR signal (*Figure 2A–D*) is a $Ca^{2+}$ release from the sarcoplasmic reticulum through ryanodine receptors, triggered by plasma membrane depolarizations above –40 mV, and thus triggered by AP in physiological conditions. Depolarization is detected by a voltage-sensor (the dihydropyridine receptor) located in the plasma membrane. This voltage-sensor derives from an L-Type $Ca^{2+}$ channel; it is voltage-dependent but not ion permeable (*Almers et al., 1981*; *Melzer et al., 1995*). Functionally linked to the ryanodine receptor at the level of the T-tubule, its activation triggers the opening of the ryanodine receptor (*Rios and Brum, 1987*; *Catterall, 1991*; *Franzini-Armstrong and Protasi, 1997*). The resulting $Ca^{2+}$ release is independent of the extra-cellular $Ca^{2+}$ (*Figure 2A*), purely voltage-dependent, and its voltage-dependency follows the typical pattern of the activation curve of a 'High Voltage-Activated' $Ca^{2+}$ channel (*Figure 2B*). Ryanodine fully blocks this $Ca^{2+}$ signal (*Figure 2C*). We hypothesized that these two distinct $Ca^{2+}$ signals, nicotinic $Ca^{2+}$ and DICR, are used by the muscle cell to detect synaptic activity and its efficiency to elicit an AP, respectively, leading to retrograde control of the presynaptic ACh release (*Figure 2* scheme). To validate this hypothesis, we examined each of these $Ca^{2+}$ dependent mechanisms in isolation and their respective impact on synaptic efficacy.

## Nicotinic $Ca^{2+}$ triggers LTP

We first examined the specific role of nicotinic $Ca^{2+}$ influx in the regulation of synaptic transmission. To trigger the nicotinic $Ca^{2+}$ influx but not the DICR in *Xenopus* muscle cells, we transiently reduced synaptic strength below AP threshold during the conditioning presynaptic burst stimulations performed in postsynaptic current-clamp. We monitored subsequent synaptic conductance changes under postsynaptic voltage-clamp during low-rate single-pulse intracellular stimulation of the neuron. We reduced the synaptic efficacy either by decreasing the synaptic conductance with low-doses of the reversible nicotinic antagonist curare (*Figure 3A–C*), or by artificially increasing the muscle cell input conductance using dynamic-clamp (*Sharp et al., 1993*; *Robinson and Kawai, 1993*; *Prinz et al., 2004*) (*Figure 3A,C*, and *Figure 3—figures supplements 1,2A*). The lowered synaptic conductance mimicked the effects of low ACh release in developing synapses. Simulation and dynamic-clamp injection of a combination of a linear and an inwardly rectifying $K^{+}$ leak conductances (Materials and methods and *Figure 3—figure supplement 1*) mimicked the increased endogenous input conductance associated with muscle cell growth. In both cases, presynaptic burst-stimulation (3 bursts of 5 pulses at 30 Hz) induced a strong, fast (within minutes) and long-term potentiation (LTP) of ePSCs (*Figure 3B,C*, and *Figure 3—figure supplement 2A*), while the transient application of curare in absence of presynaptic stimulation did not induce potentiation (*Figure 3C*, 'curare No Burst' bar). This form of LTP induced by subthreshold synaptic events depends solely on nicotinic receptor activity and therefore was insensitive to DICR blockade obtained by pre-loading the muscle cell with ryanodine (*Figure 3C*). Furthermore, when ePSPs were left intact but in presence of postsynaptic ryanodine, presynaptic burst-stimulation produced postsynaptic APs and nonetheless a strong LTP of the ePSCs (*Figure 3C*, and *Figure 3—figure supplement 2B*), while no potentiation occurred when DICR was functional (*Figure 1E* and *Figure 3C*). The LTP induced in our postsynaptic current-clamp protocols was accompanied with a strong increase of the frequency, but not amplitude, of spontaneous postsynaptic currents (sPSCs) (*Figure 3B,D*). As previously observed in voltage-clamp protocols (*Wan and Poo, 1999*), this indicates that LTP is due to an evoked ACh release increase and not to a postsynaptic receptor modulation. We found similar results in ryanodine-treated mouse muscles (*Figure 3E,F*): it is only when nicotinic channels were activated following conditioning bursts of nerve stimulations and ACh release —in the absence of DICR and contraction— that subsequent single test pulses revealed a strong LTP of the ePSPs. In contrast, as shown above, when DICR was functional and the nerve was stimulated, nicotinic ePSPs did not change and remained identical to those in the absence of nerve stimulation (*Figure 1D* and *Figure 3F*). Therefore the simultaneous occurrence of postsynaptic nicotinic receptor activation and DICR is essential

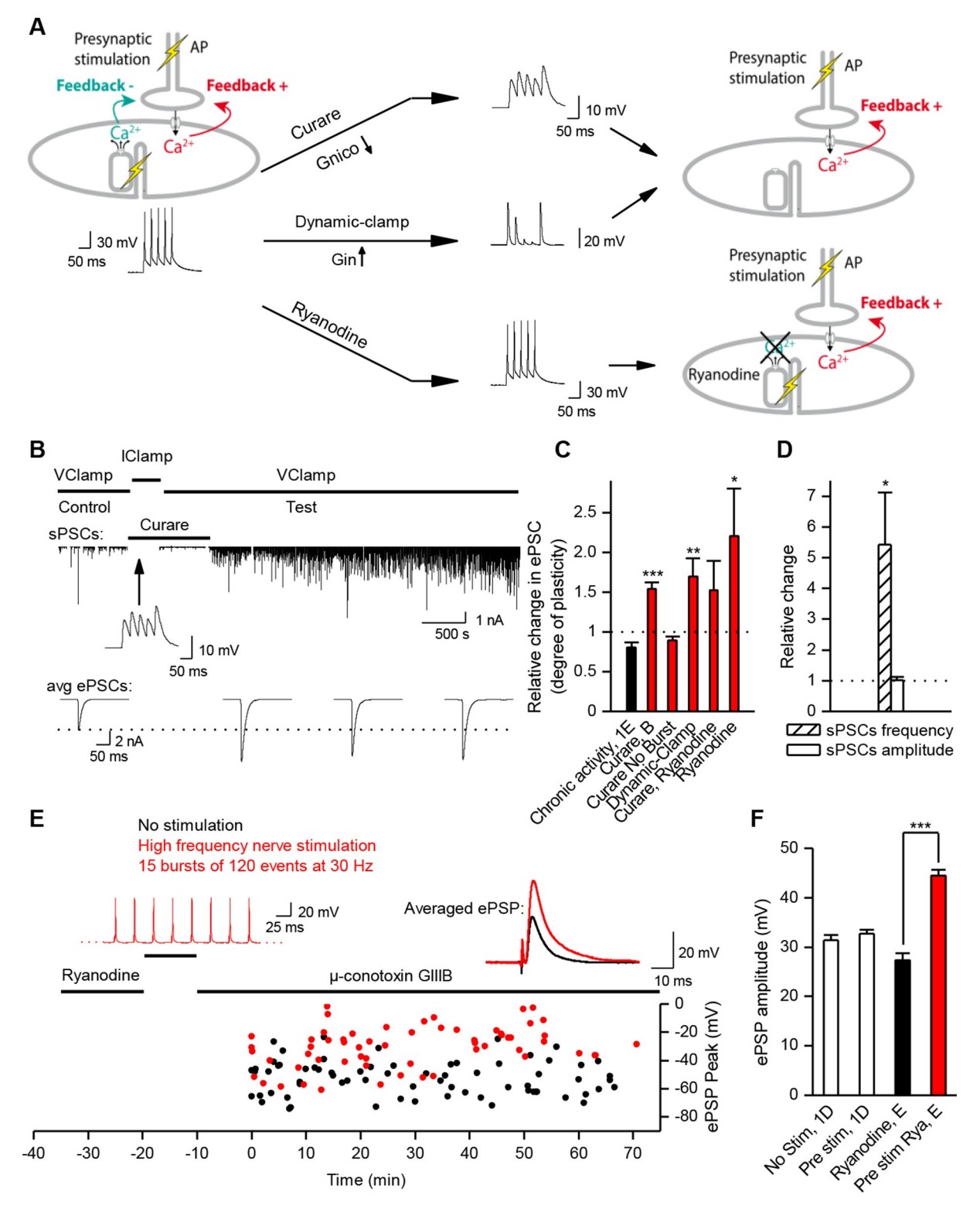

**Figure 3.** Nicotinic receptor activity induces a positive feedback on ACh release. (A) In order to obtain the nicotinic calcium signal in absence of DICR in *Xenopus*, synaptic events were transiently kept subthreshold either with curare (decreased nicotinic conductance, Gnico) or by dynamic-clamp injection of gK$^+$ leaks (increased input conductance, Gin), or left suprathreshold while DICR was blocked by ryanodine. (B) Effect of presynaptic burst stimulation and curare on spontaneous (upper trace, sPSC) and evoked synaptic currents (ePSC). ePSCs recorded under voltage-clamp were evoked at low rate (0.03 Hz) and averaged by 30–40 events (lower traces). During conditioning presynaptic stimulations (3 bursts of 5 events at 30 Hz), the

*Figure 3 continued on next page*

Figure 3 continued

postsynaptic potential was released from clamp and curare transiently applied (middle trace). Upper trace: for clarity, ePSCs were removed from the continuous trace in order to display the sPSCs only. (C), In *Xenopus*, mean ePSC relative change 30 min after control chronic bursting activity ('chronic activity', n = 5, illustrated in *Figure 1E*), 45 min after subthreshold synaptic activity ('Curare', n = 9, illustrated in B; 'Dynamic-clamp', n = 5, illustrated in *Figure 3—figure supplement 2A*), transient curare application in absence of stimulation ('curare No Burst', n = 3), sub- ('curare-ryanodine', n = 3) and suprathreshold synaptic activity ('ryanodine', n = 6, illustrated in Figure 3— figure supplement 3) in muscle cells preloaded with ryanodine. (D), Relative change in amplitude and frequency of sPSCs after potentiation in *Xenopus*. (E), In FDB mice muscles, voltage reached by ePSPs in ryanodine treated (black dots, n = 60 fibers, 2 mice), and in ryanodine treated and burst stimulated preparations (red dots, n = 70 fibers, 2 mice). (F) Mean ePSP amplitude in control ('no stim') and high frequency nerve stimulation ('Pre stim') shown in *Figure 1D*, in non-stimulated ('Ryanodine') and in high frequency stimulated ('Pre stim Rya') ryanodine treated preparations shown in E. *, p<0.05; **, p<0.01; ***, p<0.001; t-test.

The following figure supplements are available for figure 3:

**Figure supplement 1.** Decreasing muscle cell excitability by injection of artificial conductances.

**Figure supplement 2.** LTP induction with dynamic-clamp or postsynaptic ryanodine.

to the stability of functional synaptic gain. Because the regulation targets ACh release, this implies the existence of retrograde feedback onto the presynaptic compartment.

## DICR triggers LTD

In order to understand the specific role of DICR, we examined its effect on synaptic efficacy in isolation from the effect of nicotinic receptor activation. Direct stimulation of the *Xenopus* muscle cell induces postsynaptic APs and DICR, with no ACh release and no involvement of nicotinic receptors (*Figure 4A*). Conditioning bursts of suprathreshold current steps injected in the muscle cell induced a strong, fast and long-term depression (LTD) of ePSCs (*Figure 4B,C*). The average amplitude of sPSCs did not change, confirming the presynaptic locus of gain control, for negative modulations of synaptic strength just like for the positive modulations described above (*Figure 4D*). LTD induced by repetitive postsynaptic depolarizations has been previously reported (*Lo et al., 1994*; *Dan et al., 1995*). Here we show that this form of LTD relies strictly on DICR: it was insensitive to removal of external $Ca^{2+}$ (*Figure 4C* and *Figure 4—figure supplement 1A*), and blocked by postsynaptic preloading with ryanodine (*Figure 4C* and *Figure 4—figure supplement 1B*). In adult mouse nerve-muscle preparations, external electrical stimulation elicits an AP in both the nerve and the muscle fibers, a situation in which both DICR and nicotinic $Ca^{2+}$ influx occur. As expected, in the presence of external $Ca^{2+}$, the external burst stimulation did not change the average ePSP (*Figure 4E, F*) compared to nerve stimulation only (*Figure 1D* and *Figure 4F*). When the external $Ca^{2+}$ was transiently removed, the external stimulation still elicited APs in both the nerve and the muscle, but neither the ACh release nor its associated postsynaptic nicotinic $Ca^{2+}$ influx. In these conditions, the external stimulation induced DICR alone and resulted in a strong LTD of the ePSPs (*Figure 4E,F*). This form of depression is reversible, since subsequent burst stimulation of the nerve induced a rapid potentiation of the ePSPs (*Figure 4—figure supplement 2*).

## Plasticity orientation rule is homeostatic

In the previous experiments, we used the extreme cases of either fully subthreshold synaptic events or of postsynaptic APs in the absence of presynaptic neuron activity, to emphasize the roles of the nicotinic calcium and the DICR signals respectively. However, during normal synaptic activity the two calcium signals are combined and a unique plasticity orientation rule, reflecting the interaction between the potentiating and depressing synaptic processes, should be possible to define in terms of synaptic efficacy.

Patch-clamp on *Xenopus* cells in culture allows the examining of changes in the synaptic strength in individual neuromuscular synapses, and the assessing of synaptic conductance. In *Figure 1E*, we observed in non-treated *Xenopus* synapses that 30 min of chronic burst stimulation of the motor neuron did not drastically change the synaptic gain, but nevertheless induced a slight depression of the averaged ePSCs. Next, we set to closely examine these small synaptic changes in individual synapses. Given the variety of average synaptic conductance among neuromuscular synapses

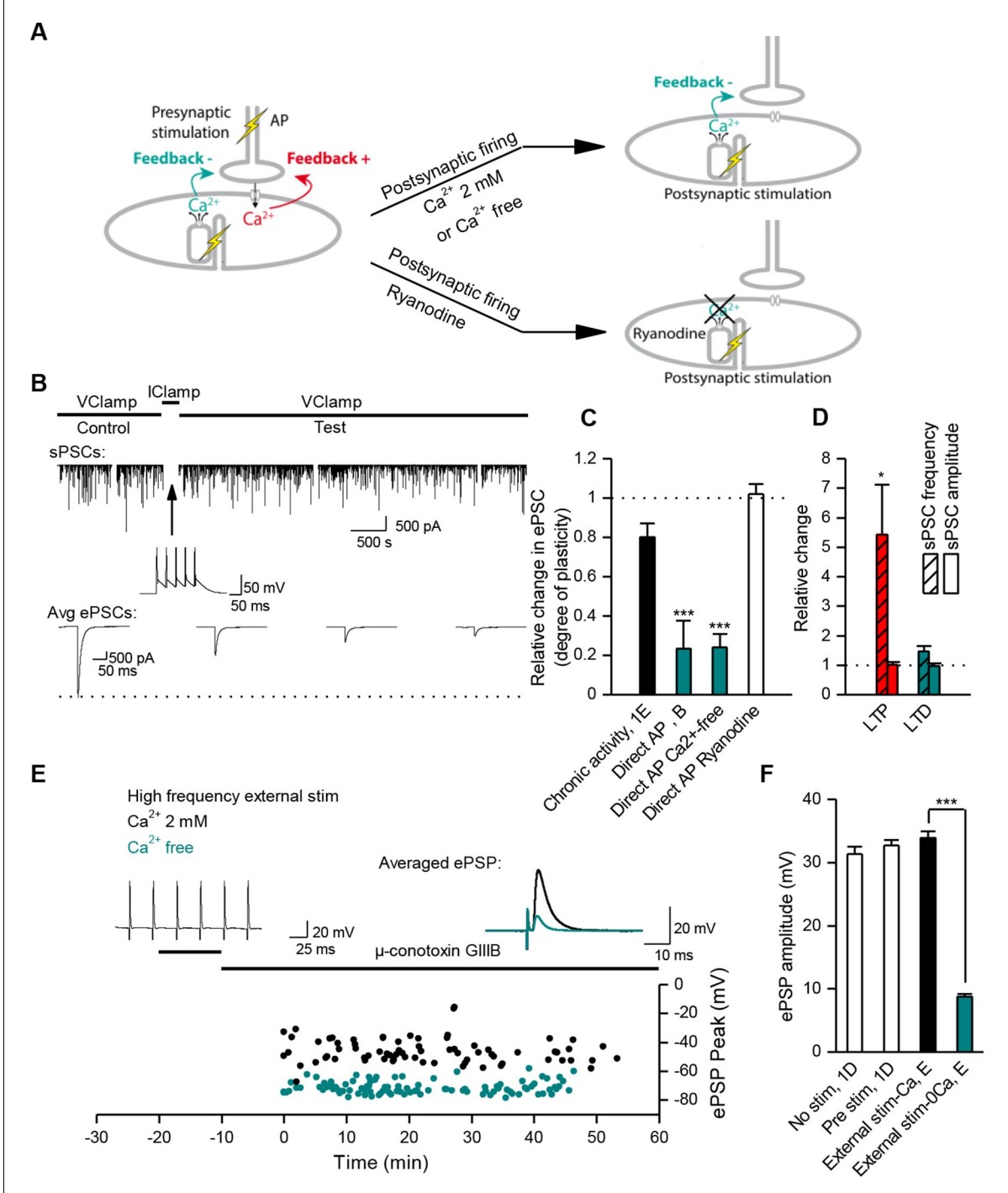

**Figure 4.** Muscle DICR induces a negative feedback on ACh release. (A) In order to trigger the DICR signal in absence of nicotinic $Ca^{2+}$ influx in *Xenopus*, postsynaptic APs were directly induced in the muscle cell with brief current steps, without presynaptic stimulation, in presence or absence of external $Ca^{2+}$ (upper right scheme) and with ryanodine that blocks the DICR (bottom right scheme). (B) Effect of selective postsynaptic firing on synaptic currents. Same layout as in *Figure 3B*. The middle trace shows the muscle APs firing in response to positive current steps injection. (C) In *Xenopus*, mean ePSC relative change 45 min after control chronic bursting synaptic activity ('chronic activity', n = 5, see 1E), direct triggering of the muscle APs shown in B ('Direct AP', n = 5), APs triggered in a $Ca^{2+}$-free medium ('Direct AP $Ca^{2+}$-free', n = 6, illustrated in *Figure 4—figure supplement 1*), APs triggered in muscle cells loaded with ryanodine ('Direct AP Ryanodine', n = 3, illustrated in *Figure 4—figure supplement 2*). (D), Relative change in sPSCs amplitude and frequency in *Xenopus* after potentiation (red) or depression (green). (E), In FDB mouse muscles, voltage reached by ePSPs in externally stimulated preparations (15 bursts of 120 events at 30 Hz, 10 min) in presence (black dots, n = 75 fibers, 2 mice) and in absence of external $Ca^{2+}$ (green dots, n = 120 fibers, 2 mice). (F), Mean ePSP amplitude in control ('no stim') and high frequency nerve stimulation ('Pre stim') shown in

*Figure 4 continued on next page*

*Figure 4 continued*

**Figure 1D**, in externally stimulated preparations shown in E in presence ('External stim-Ca$^{2+}$') and absence of external Ca$^{2+}$ ('External stim-0Ca$^{2+}$'). ***, p<0.001.
The following figure supplements are available for figure 4:

**Figure supplement 1.** External calcium-independent LTD and blockade by postsynaptic ryanodine.
**Figure supplement 2.** Recovery from LTD.

(*Figure 1C*, Y axis), and its strong correlation with the muscle input conductance (*Figure 1C*), the range of the ratio between the synaptic and the input conductances 'Gsyn/Gin' is much tighter than the synaptic conductance range. We reasoned that because this ratio, and not the synaptic conductance alone, determines the synaptic efficacy (ePSP amplitude), it should be the appropriate synaptic parameter targeted by the homeostatic process. Therefore, we recorded the ePSCs under brief test periods of postsynaptic voltage-clamp, before and after 20–30 min of chronic burst stimulation of the motor-neuron under postsynaptic current-clamp, and normalized these averaged ePSCs by the input conductance of the muscle cell. *Figure 5A* (black dots) shows that the slight depression was not homogenous among neuromuscular synapses. The degree of depression seems to depend on the distance of the initial Gsyn/Gin ratio from the mean ratio obtained after chronic activity (*Figure 5A*, dashed line). Consequently, standard deviation from the mean is reduced after chronic activity (*Figure 5A*, inset). This synaptic depression behavior can be interpreted as the convergence of the Gsyn/Gin ratios towards the set point of the homeostasis.

In order to test whether synapses also converge if the initial Gsyn/Gin ratio is below the set point, we applied the same chronic activity in the continuous presence of low doses of curare (0.1–2 µM). At these concentrations, the bursts of evoked synaptic events are still composed of a majority of suprathreshold events, combining both potentiating and depressing processes. *Figure 5A* (red dots) shows that, again, the degree of plasticity (potentiation in this case) depends on the distance of the initial Gsyn/Gin ratio to the set point.

The plasticity orientation rule may then be expressed in a way showing its homeostatic character. We plotted the degrees of plasticity (relative change in the Gsyn/Gin ratio) in non-treated (*Figure 5B*, black dots) and curare treated (*Figure 5B*, red dots) synapses shown in *Figure 5A*, as a function of the difference between the initial Gsyn/Gin ratios and the mean Gsyn/Gin ratio after chronic activity (distance to the set point). In a perfect homeostatic process, with an attractor set point, the degree of plasticity can be expressed as a function of the distance to the set point:

$$\frac{\left(\frac{Gsyn}{Gin}\right)_{after}}{\left(\frac{Gsyn}{Gin}\right)_{before}} = 1 - \frac{1}{1 + \frac{set\ point}{\left(\frac{Gsyn}{Gin}\right)_{before} - set\ point}}$$

The solid line in *Figure 5B* shows this theoretical relationship with a set point of 2.36, equal to the mean Gsyn/Gin ratio obtained after chronic activity. The overlap between the data points and the theoretical relationship shows the homeostatic function of the neuromuscular plasticity.

In order to place the potentiation we obtained with ryanodine (*Figure 3C*, 'ryanodine' bar) in regard of this homeostatic rule, we normalized the averaged synaptic conductances by the input conductance and added the data to the *Figure 5A* (green dots). The ryanodine-treated synapses did not converge towards a set point. The averaged Gsyn/Gin ratios after burst activity were above the set point, and the standard deviations from the mean were increased. These data suggest that the ryanodine receptors-dependent calcium signal participates to the stabilization of the synaptic efficacy at the set point.

In adult mouse neuromuscular synapses, an apparent Gsyn/Gin ratio can be extrapolated from the ePSPs. If we assume a linear passive leak, the Gsyn/Gin ratio can be expressed as:

$\frac{Gsyn}{Gin} = \frac{-80 - Vp}{Vp}$ with Vp the peak membrane potential reached by the ePSP and 0 and -80 mV the reversal potentials of the synaptic and the leak currents respectively. These ratios are only apparent and underestimated given the effect of distance between the recording site and the neuromuscular junction. *Figure 5C* shows the apparent Gsyn/Gin ratio in non-stimulated and in burst-stimulated

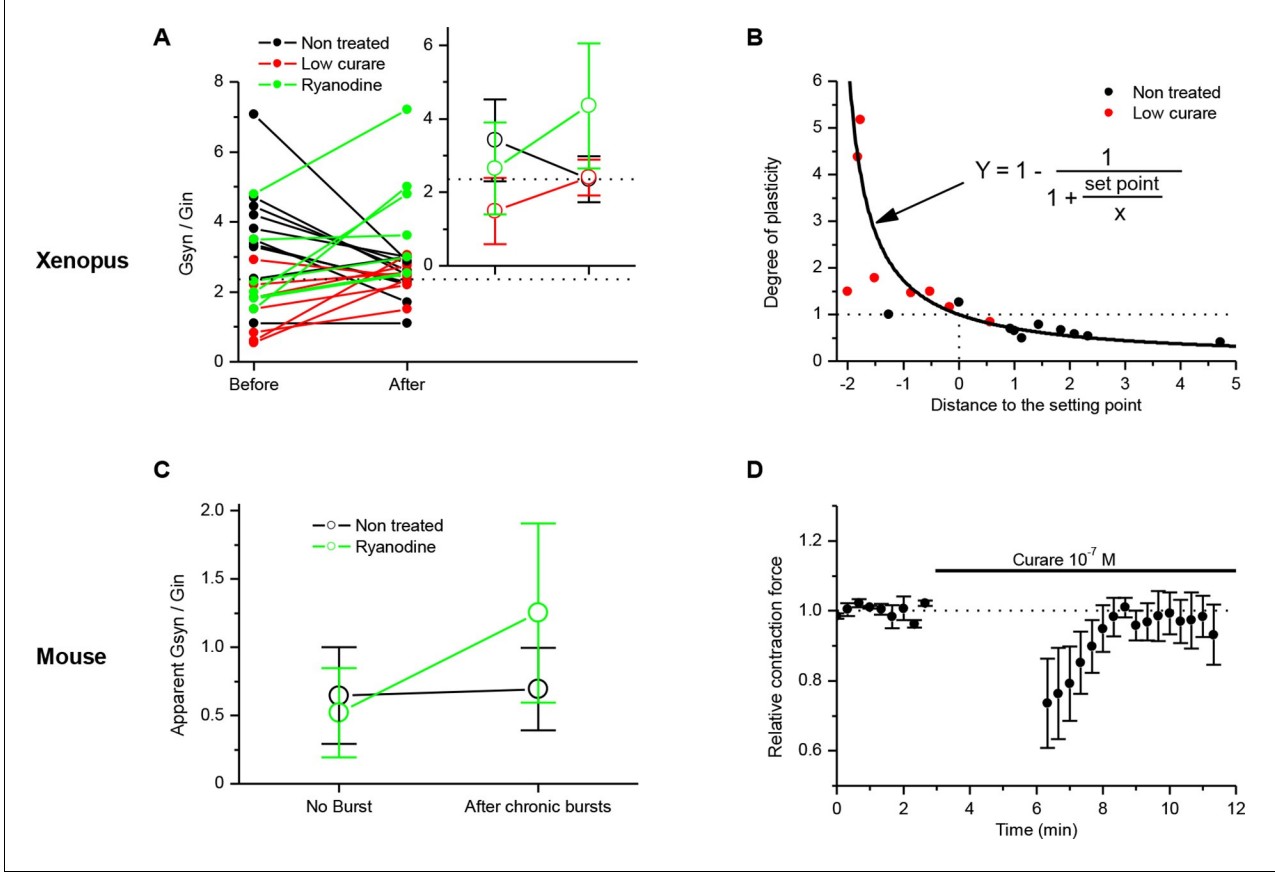

**Figure 5.** Homeostatic control of the synaptic efficacy. (**A**) In *Xenopus*, ratios between averaged synaptic conductance ('Gsyn', calculated from 30–40 ePSCs) and muscle cell input conductance ('Gin') before and after 20–30 min of chronic burst stimulation of the motor neuron (burst of 20–60 events at a 20–30 Hz frequency, every 30–40 s) under postsynaptic current-clamp in non-treated (black dots) and low curare-treated (red dots) synapses. Green dots represent the Gsyn/Gin ratio before and after 1–3 bursts of 5 presynaptic stimulations at 30 Hz (green dots) in ryanodine loaded muscle cells (same data than in *Figure 3C*, ryanodine bar). Inset shows the mean ± standard deviation of the Gsyn/Gin ratios in the three conditions. The dotted lines show the averaged Gsyn/Gin ratio after chronic activity in non-treated synapses. (**B**) Degree of plasticity (relative change in Gsyn/Gin ratio) shown in A expressed as a function of the difference between the initial individual Gsyn/Gin ratio and the averaged ratio after chronic burst activity ('Distance to the set point'), in non-treated (black dots) and curare-treated (red dots) synapses. The solid line shows the theoretical homeostatic relationship between plasticity and the distance to a set point of 2.36, calculated as the mean Gsyn/Gin ratio after chronic activity in non-treated synapses. (**C**) Apparent averaged Gsyn/Gin ratios calculated in mouse FDB muscles from the data of *Figure 3E*, in non-stimulated and non-treated synapses (no burst, black dot), in burst-stimulated and non-treated synapses (after chronic bursts, black dot), in non-stimulated and ryanodine-treated synapses (no burst, green dot) and in burst-stimulated and ryanodine-treated synapses (after chronic bursts, green dot). Dots represent the mean ± Standard Deviation. (**D**) Relative change of contraction force during 2s-30Hz bursts of nerve stimulations in mouse soleus muscles (n=4), before and during exposure to a low dose (0.1 μM) of curare.

preparations, both for non-treated (black dots) and ryanodine-treated (green dots) preparations (same data as in *Figure 1D* and *Figure 3D–E*). These data in mouse (*Figure 5C*) confirmed the results seen in *Xenopus* (*Figure 5A*).

Finally, in mouse, we applied a continuous low dose of curare (0.1 μM) on soleus muscles, together with chronic burst stimulation of the nerve, in order to show that the homeostatic processes are also able to compensate for a decrease in the postsynaptic sensitivity to the neurotransmitter. At this curare concentration, the synaptic activity is still composed of a majority of suprathreshold events, combining both nicotinic and DICR calcium signals. We took advantage of the fact that the muscle contraction force integrates the synaptic efficacy over all the fibers of the muscle and all the synaptic events of the bursts, and as such is an indicator of even small changes in the synaptic gain. *Figure 5D* shows the 26% decrease in the contraction force due to low curare, followed by progressive recovery of the force back to a normal level in response to subsequent nerve burst stimulations.

We could thus directly observe the functional effect of the synaptic homeostasis mechanism we describe.

## Discussion

We have demonstrated in two evolutionarily distant vertebrate species that the skeletal muscle cell constantly senses the synaptic events and their ability to trigger postsynaptic action-potentials. Two distinct calcium signals report the pre- and postsynaptic activity. Synaptic events are detected through the nicotinic $Ca^{2+}$ influx and trigger a positive retrograde feedback that targets presynaptic neurotransmitter release. Postsynaptic APs are detected through the sarcoplasmic reticulum DICR signal and trigger a negative retrograde feedback. In *Xenopus*, convergence of the synaptic efficacy —here captured by the ratio between the synaptic and the input conductances— towards an attractor set point located in the suprathreshold range of synaptic efficacy suggests that a dynamic balance between the potentiating and depressing synaptic processes ensures homeostasis of the neuromuscular transmission. This equilibrium automatically sets the neurotransmitter release to a level matching the postsynaptic excitability. Obviously, ACh release is not the only actor of synaptic efficacy, nicotinic receptors density and muscle input conductance being also under control. However, the advantages of the calcium sensors-based plasticity of presynaptic release are its dynamical aspect and its fast kinetics, confirming in vertebrates that rapid adjustment of the neurotransmitter release is a primary variable in the homeostatic control of neuromuscular synaptic efficacy (*Plomp et al., 1992*; *Petersen et al., 1997*; *Davis et al., 1998*; *DiAntonio et al., 1999*; *Paradis et al., 2001*; *Frank et al., 2006*; *Davis and Müller, 2015*).

This mechanism may primarily explain the ubiquity of the unity neuromuscular synaptic gain across different muscles and species, and is likely to be involved in several synaptic regulation processes that occur throughout the life of the neuromuscular synapse. In particular, it could be involved in the development and stabilization of forming synapses, in the heterosynaptic depression (*Lo and Poo, 1991*; *1994*) which takes part in the important developmental phase of synaptic selection leading to the characteristic mono-innervation (*Gouzé et al., 1983*; *Sanes and Lichtman, 1999*; *Witzemann, 2006*), and it could counterbalance modulations of muscle excitability associated with growth and physical exercise. This homeostatic control might also be involved in the quantal content increase which partially compensates for the reduced number of postsynaptic receptors at myasthenic human end-plate (*Cull-Candy et al., 1979*; *1980*), or in rat diaphragm muscle treated with the nicotinic antagonist α-bungarotoxin, mimicking myasthenia gravis syndrome (*Plomp et al., 1992*).

### A novel plasticity orientation rule

When selectively triggered, the nicotinic $Ca^{2+}$ influx and the DICR signal induce a form of LTP and LTD respectively. The causal links between the nicotinic $Ca^{2+}$ influx and the synaptic events, and between the DICR signal and the muscle APs, allow the establishment of an orientation rule for neuromuscular synaptic plasticity: synaptic events-induced LTP *versus* muscle AP-induced LTD. This orientation rule finds its equilibrium in the suprathreshold range of the synaptic strength and provides the specific 'homeostatic' function to this plasticity. This neuromuscular plasticity differs from the other forms of long-term plasticity, and in particular should not be confounded with anti-Hebbian plasticity (*Kullmann and Lamsa, 2008*; *Roberts and Leen, 2010*). Anti-Hebbian plasticity underlies synaptic strength modulations while the present homeostatic mechanism maintains the stability of synaptic efficacy.

Modulation of synaptic transmission has been extensively studied in vitro at the *Xenopus* neuromuscular synapse by Poo group (*Fu and Poo, 1991*; *Lo and Poo, 1991*; *1994*; *Lohof et al., 1993*; *Lo et al., 1994*; *Dan et al., 1995*; *Cash et al., 1996*; *Wang and Poo, 1997*; *Wan and Poo, 1999*). Our work, partly done on the same experimental model, suggests that a number of results obtained by Poo group might be interpreted in the more specific framework of homeostatic plasticity.

In vertebrates, neurotrophic factors such as brain-derived neurotrophic factor, neurotrophin 3 and neurotrophin 4, are trans-synaptic factors considered candidates to mediate presynaptic release modulations in many forms of synaptic plasticity (*Schinder and Poo, 2000*; *Poo, 2001*). Skeletal muscle synthesize and release neurotrophins in an activity-dependent manner (*Funakoshi et al., 1995*; *Wang and Poo, 1997*; *Xie et al., 1997*), and motor nerve terminals contain the receptors

tyrosine kinase TrkB and C (*Henderson et al., 1993*; *Koliatsos et al., 1993*; *Wong et al., 1993*; *Yan et al., 1993*). At the *Xenopus* neuromuscular synapse in vitro, Poo group has shown that the exogenous application (*Lohof et al., 1993*; *Wang and Poo, 1997*) of these factors reproduces the synaptic LTP induced in the present work with subthreshold synaptic events or postsynaptic ryanodine treatment. Finally, upregulation of the ACh evoked release in α-bungarotoxin treated rat was markedly reduced by inhibition of the tyrosin kinase receptors of the neurotrophins (*Plomp and Molenaar, 1996*). Therefore, neurotrophic factors should be considered for future investigations to determine whether they can be retrograde factors mediating homeostatic plasticity at the neuromuscular synapse.

## Postsynaptic calcium signaling

The diversity of the effects of the calcium signaling comes from the variety of its temporal patterns, amplitude, and location. The two antagonist calcium signals that we demonstrated here to be involved in the orientation of the neuromuscular plasticity exhibit different temporal patterns and amplitudes. The AP-associated DICR signal exhibits fast transient large calcium concentration elevations for each AP. The kinetics of the nicotinic calcium build-up is much slower and does not contain transients during repetitive nicotinic receptor stimulations. Its amplitude under voltage-clamp is low compare to the DICR signal (*Figure 2F*), and even lower under current-clamp given the reduced driving-force for the calcium ion due to the postsynaptic depolarization. This dichotomy between fast large transient and slow low calcium build-up is known to trigger selectively, *via* calmodulin, the CAMKII kinase and the calcineurin phosphatase respectively. This mechanism is considered in the central nervous system to implement Hebbian plasticity (*Malenka et al., 1989*; *Malinow et al., 1989*; *Mulkey et al., 1993*; *1994*) by modulating the conductance and number of the postsynaptic receptors (*Barria et al., 1997*; *Mammen et al., 1997*; *Beattie et al., 2000*). The CAMKII signaling pathway was also shown to be required for normal 'presynaptic homeostasis' in *Drosophila* (*Haghighi et al., 2003*). Furthermore, *Wan and Poo 1999* showed at the *Xenopus* synapse in vitro that induction of LTP and LTD can be blocked by pre-loading the muscle cell with peptide inhibitors of calcineurin and CAMKII respectively. As noted by *Wan and Poo 1999*, this situation is opposite to the central synapses, where calcineurin is associated to depression and CAMKII to potentiation. Our results suggest that the low-calcium nicotinic signal and the calcineurin activation have a possible causal link with the detection of the synaptic events and the synaptic potentiation, while the high-calcium DICR signal and the CAMKII activation are associated with the detection of the postsynaptic APs and the synaptic depression.

## A push-pull mechanism for homeostatic plasticity

The convergence of the synaptic efficacy towards a set point (*Figure 5*) suggests that the potentiating and depressing synaptic processes balance each other in the suprathreshold range of the synaptic strength. These observations strongly suggest that the set point of the homeostatic plasticity is sustained by a push-pull mechanism. Our results, however, do not determine the stages where this push-pull mechanism could operate in the causal chain linking the evaluation of synaptic efficacy to the presynaptic modulation. In particular, our findings do not necessarily imply that two independent antagonist trans-synaptic retrograde factors balance their effects at the presynaptic level. The coexistence of potentiating and depressing trans-synaptic factors is a possibility among others. The increased and decreased secretion of a single positive retrograde factor —like for example endostatin in *Drosophila* (*Wang et al., 2014*) and neurotrophins in vertebrates— could also bi-directionally regulate neurotransmitter release. In this case, it could be hypothesized that a push-pull mechanism operates downstream of the calcium signaling, for example at the level of the calcineurin/CAMKII balance, ultimately determining the secretion level of a single factor. Therefore, the use of blue and red arrows in *Figure 2* is intended as a schematic representation of the retrograde mechanisms that can bi-directionally change neurotransmitter release, without presuming the existence of two distinct retrograde factors.

## Comparison with the *Drosophila* model

Most of what we know about synaptic homeostasis stems from experimental studies at the *Drosophila* larva neuromuscular junction. Given the numerous differences between *Drosophila* and

vertebrates the comparison between these systems is not straightforward. Action potentials and the associated DICR signal in vertebrates being absent in *Drosophila* muscle cells (*Hong and Ganetzky, 1994*), the homeostatic mechanism we propose here cannot be directly transposed to *Drosophila*. Since no orthologue of the tyrosine kinase receptors are found in *Drosophila* (*Frank, 2014*), the neurotrophin hypothesis in vertebrates cannot either be applied to *Drosophila*. However, different actors could play similar roles in both systems. The calcium permeability of glutamate receptors in *Drosophila* could have a similar role than the calcium permeability of nicotinic receptors in vertebrates. The low-voltage activated calcium channels activated by the ePSPs and the 'calcium-induced calcium release' signal responsible for the excitation-contraction coupling in *Drosophila* (*Peron et al., 2009*) could be the orthologue of the DICR signal in vertebrates. Finally, endostatin in *Drosophila* (*Wang et al., 2014*) may play the role of neurotrophins in vertebrates.

In *Drosophila*, *Frank et al. (2006)* found that spontaneous glutamate release was sufficient to induce a rapid potentiation in the absence of nerve activity, while we had to use a 30Hz motor command to induce rapid potentiation in vertebrates. In order to establish the *Figure 5A*, cells were incubated in curare in absence of neurons spikes during 30–60 min before recording (red dots). Therefore, the spontaneous ACh release did not restore the normal Gsyn/Gin ratio (found in non-treated synapses) before chronic bursts were applied. A possible explanation of this difference between the two systems is the high frequency of miniatures in *Drosophila* (10–20 Hz) (*Frank et al., 2006*), while the average frequency of spontaneous release in the *Xenopus* cells culture was 100 fold lower. In vertebrates, evoked activity responsible for muscular tonus and voluntary motions represents most of the synaptic activity, and the 20–30 Hz frequencies we used in our experiments are in the normal range for a motor command (*Gorassini et al., 2000*). Therefore, the evoked activity in vertebrate is more likely able to rapidly mobilize the homeostatic machinery than the spontaneous miniatures.

## NMJ as a model for homeostasis in the central nervous system?

In the central nervous system, homeostatic forms of synaptic plasticity have been proposed to restrain the mean level of neuronal activity within a physiological regime, and to maintain the stability of recurrent network activity challenged by associative plasticity. Because of the robustness of its synaptic transmission, the neuromuscular junction is considered as a model of homeostasis. However, the neuromuscular junction being the end effector of the motor network, its main function is the faithful translation of the motor command into muscle activity. The mean level of muscle activity is not involved in any recurrent network that requires stability, and therefore does not need to be controlled. While the synaptic strength is the adjustment variable for the control of the mean level of neuronal activity in the CNS, at the NMJ it is the efficacy of synaptic transmission itself that must be maintained constant for reliable relay of the motor command. Therefore, homeostatic plasticity could be regarded as a functionally different phenomenon at the NMJ than in the CNS. Nevertheless, the plasticity orientation rule described in the present work matches a synaptic 'relay' function. Despite differences in the nature of the activity sensors, other 'relay' synapses in the CNS, such as those involved in sensory inputs to the thalamus (*Guido, 2008*), might share with the NMJ a comparable homeostatic control based on pre- and post- synaptic activity detection.

## Material and methods

Animal care followed the European Union regulations (OJ of EC L358/1 18 December 1986), and the European directive 2010/63/UE.

### Primary cell culture

Myotomal and spinal tissues from 1-day-old *Xenopus laevis* embryos (stages 23 to 25) were mechanically dissociated using a $Ca^{2+}$- and $Mg^{2+}$-free medium of the following composition (in mM): 115 NaCl, 2.6 KCl, 0.4 EDTA, 10 HEPES (pH = 7.6). Cells were directly plated in a plastic recording chamber, and grown at 19°C for 12 hr prior to the experiments. The culture medium consisted of 50% Leibovitz L-15 medium (Gibco, Invitrogen Corp., Cergy-Pontoise, France), 1% fetal calf serum, 1% antibiotic mixture (*ibid.,* final concentration: 100 units/mL penicillin G and 100 µg/mL streptomycin), and 48% physiological solution of the following composition (in mM): 113 NaCl, 2 KCl, 0.7 $CaCl_2$, 5 HEPES (pH=7.8).

## Patch-clamp recordings

Perforated patch-clamp recordings, to preserve the cell integrity, were performed at room temperature (20–22°C). Pipettes were made from borosilicate glass (Clark Electromedical Instruments, Reading, England) and pulled on a P-1000 puller (Sutter Instrument Company, Novato, CA, U.S.A.). Patch electrodes had a resistance of 2–3 MΩ when filled with internal physiological solution. Membrane currents and potential were recorded using Axopatch200B patch-clamp amplifiers (Axon Instruments, Union City, CA). Access resistances were compensated at 80%. Myocyte membrane currents were filtered with an integrated low-pass Bessel filter at 2 kHz. The filtered signals were digitized by a 12 bit A/D converter (Digidata 1200B, *ibid.*) and stored using pCLAMP 8 software (*ibid.*). Recordings were analyzed using the Origin 7 software (OriginLab Corp., Northampton, MA). Motor neurons were current-clamped through an amphotericin-perforated membrane patch (400 pA, 3 ms current step to induce the presynaptic AP); the intra-pipette solution had the following composition (in mM): 1 NaCl, 140 K-gluconate, 1 MgCl2, 10 HEPES (pH=7.2), and amphotericin-B at 300 µg/ml. Myocytes were either voltage- or current-clamped, through an amphotericin-perforated membrane patch, using the following intra-pipette composition (in mM): 1 NaCl, 20 KCl, 125 K-gluconate, 1 MgCl2, 10 HEPES (pH=7.2), and amphotericin-B at 300 µg/ml. The external solution had the following composition (in mM): 140 NaCl, 3 KCl, 1 MgCl2, 2 CaCl2, 10 HEPES (pH=7.4).

## Iontophoretic ACh applications

Pipettes made with borosilicate glass had a resistance of 80–120 MOhm when filled with 0.5 to 1 M ACh-Cl. $ACh^+$ efflux was induced by positive current steps, using a home-made constant current generator.

## Fluorescence

*Xenopus* muscle cells were loaded with the calcium indicator in whole-cell patch-clamp configuration for 10 min with an intra-pipette medium of the following composition (in mM): 110 KCl, 1 NaCl, 2 $Mg^{2+}$, 10 HEPES (pH 7.2), and 0.2 Fluo4-pentapotassium salt. The patch pipette was then removed and a perforated patch was performed as described above. Fluorescence intensity was quantified with an Olympus photomultiplier (forming part of the OSP system, Olympus, Japan), and the tension signal was digitized with the Digidata converter. After background fluorescence subtraction, signals were normalized according to the baseline fluorescence.

## Nicotinic ionic permeability

The reversal potential of the nicotinic current induced by ionophoretic acetylcholine application was measured under voltage-clamp in solutions of various ionic compositions (*Figure 2—figure supplement 1*). The following equation, derived from the Goldman-Hodgkin-Katz flux equation(*Goldman, 1943*; *Hodgkin and Katz, 1949*), was used to calculate the permeability ratios:

$$V_{rev} = \frac{RT}{F}\left[\frac{0.75 \times P_K[K]_o + 0.75 \times P_{Na}[Na]_o + 0.25 \times 4P_{Ca}[Ca]_o + 0.25 \times 4PMg[Mg]_o}{0.75 \times P_K[K]_i + 0.75 \times P_{Na} \times [Na]_i + 0.25 \times Mg[Mg]_i}\right]$$

In this equation, the terms in square brackets are ion concentrations, $P_S$ is the permeability of the S ion species, T is the absolute temperature, R and F are the gas and Faraday constants, and i and o are the intra- and extracellular compartments, respectively. Factors represent the ionic activity coefficients.

## Dynamic-clamp

The dynamic-clamp technique (*Sharp et al., 1993*; *Robinson and Kawai, 1993*) was used to inject computer-generated conductances in *Xenopus* muscle cells. Dynamic-clamp experiments were run using the hybrid RT-NEURON environment (*Le Franc et al., 2001*; *Sadoc et al., 2009*), a modified version of NEURON (*Hines and Carnevale, 1997*) running under the Windows operating system (Microsoft Corp., Redmond, Washington), augmented with the capacity of simulating models in real time, synchronized with the intracellular recording. A PCI DSP board with 16 bit A/D-D/A converters (Innovative Integration, SimiValley) was used to input the membrane potential into the equations of the model, and to output the current to be injected into the cell with a time resolution of 0.1 ms. Passive $K^+$ leak and $K^+$ inward rectifying (Kir) conductances were injected into the muscle cell to

simulate an increase in the input conductance (*Figure 3—figure supplement 1*). The passive K⁺ leak was expressed in the form: Ileak = gleak x (V-Eleak). Eleak was set to -80 mV, and gleak to 5–15 nS. The Kir conductance was expressed in the form: IKir =gmax x m x (V – E$_K$)

where the maximal conductance gmax was set to 20–50 nS, and changes with time of the gating variable m were calculated by solving the differential equation:

$$\frac{dm}{dt} = \frac{m_\infty - m}{\tau_m}$$

$$m(t) = m_\infty - (m_\infty - m_0) \times \left( -\frac{t}{\tau_m} \right)$$

where fitting voltage-clamp recordings of the real isolated Kir current of *Xenopus* muscle cells gave and $m_{\infty(v)} = \frac{1}{1+exp(-0.074 \times (E_K - V))}$ and $\tau_m = 0.2ms$

## Measurement of Xenopus muscle cell contraction

The 'contraction' trace shown in *Figure 2A* represents a qualitative measurement of the contraction of a muscle cell in culture. In addition to the membrane potential recording pipette, a second patch pipette vertically approached the membrane cell until a slight increase in the pipette electrical resistance became visible. The pipette was then held in that position, and increase in the cell thickness accompanying contraction was monitored through further variations of the pipette resistance.

## Adult mouse nerve-muscle preparations

3- to 5-month-old Swiss mice were anesthetized with isoflurane, and cervical dislocated. Dissections were performed within 15 min in an oxygenated Ringer solution of the following composition (in mM): 145 NaCl, 3 KCl, 2 CaCl2, 1 MgCl2, 10 HEPES (pH 7.4) and 11 glucose.

## Floating electrode recordings

Intracellular recordings (*Figure 1—figure supplement 1*) were performed at 34°C, in the oxygenated Ringer solution defined above. Sharp pipettes were made from borosilicate glass (Clark Electromedical Instruments), pulled on a P-1000 puller (Sutter Instrument Company), and had a resistance of 40–60 MΩ when filled with a KCl 3 M solution. The filled pipette tip was cut at the limit of the pulled zone, and used as electrode. The chlorided end of a 10 cm long, 50 µm diameter, silver wire was introduced inside this electrode, and plugged by its opposite end to the headstage of an Axoclamp 2A amplifier (Axon Instruments), where it hung loosely above the muscle. A drop of mineral oil was added at the back of the intra-pipette medium to avoid evaporation. The pipette pendulum was vertically manipulated to enter the muscle cells, and its flexibility allowed stable membrane potential recordings in contracting muscles. Nerve APs were induced with 6 V, 30 µs voltage steps in a suction pipette. 2 µM of µ-conotoxin GIIIB (10 min) was used to isolate the ePSPs. Data acquisition and analysis were made as above.

Under µ-conotoxin the amplitudes of the recorded ePSPs depend on the distance between the recording site and the synaptic region. With the floating electrode method, the placement of the electrode is not precisely controlled. We limited the distance effect by using the FDB mouse muscle, where cells are shorter (300 µm) than the muscle length (1 cm). Therefore, blind placement of the electrode can be performed on this muscle, with a maximal 150 µm distance between the recording site and the end plate. The specific cells size and organization in FDB limit the distance effect in our recordings, but this effect is presumably responsible for an underestimation of the ePSPs amplitudes and for most of the variability between the recorded cells. Despite the underestimation of the ePSPs amplitudes, the recorded amplitudes were above the -63 mV AP threshold determined in muscle cells of rat fast and slow muscles (*Wood and Slater, 1995*).

## Products

The salts composing the media used for electrophysiological recordings, EGTA used to obtain Ca²⁺-free media, and Amphotericin B used for perforated patch-clamp were obtained from Sigma-Aldrich (Sigma-Aldrich, Saint Quentin Fallavier, France). Ryanodine used in some experiments was 'Ryanodine fractions' from Latoxan (Latoxan, Valence, France). The µ-conotoxin GIIIB used to block mouse

muscle action-potentials was obtained from Alomone Labs (Alomone Labs, Jerusalem, Israel). The $Ca^{2+}$ dye 'Fluo4-pentapotassium salt' was obtained from Molecular Probes (Molecular Probes, Eugene, Oregon).

## Acknowledgements

We thank Jean-Pierre Changeux, David McCormick and Yves Frégnac for critical discussions about the manuscript, Zuzanna Piwkowska, Evan Harrell and Francesca Barbieri for proof reading, and Patrick Parra and Jean-Yves Tiercelin for shaping the mechanical parts of the experimental setups.

GO, GB and TB were supported by the 'Centre National de la Recherche Scientifique' (CNRS). GO and GB were supported by a grant from 'Association Française contre les Myopathies'. GO and TB were supported by grants from 'Neuropôle de Recherche Francilien' and 'Fondation pour la Recherche Médicale'.

## Additional information

### Funding

| Funder | Author |
| --- | --- |
| Centre National de la Recherche Scientifique | Gilles Ouanounou<br>Gérard Baux<br>Thierry Bal |
| Association Française contre les Myopathies | Gilles Ouanounou<br>Gérard Baux |
| Fondation pour la Recherche Médicale | Gilles Ouanounou<br>Thierry Bal |
| Neuropole de recherche francilien | Gilles Ouanounou<br>Thierry Bal |

The funders had no role in study design, data collection and interpretation, or the decision to submit the work for publication.

### Author contributions

GO, Designed the research, Performed experiments, Analysis and interpretation of data, Wrote the paper ; GB, Contributed with critical discussion on the manuscript, Drafting or revising the article; TB, Provided the dynamic-clamp technique, Contributed with critical discussion on the manuscript, Drafting or revising the article

### Author ORCIDs

Gilles Ouanounou, http://orcid.org/0000-0001-5658-7005

### Ethics

Animal experimentation: Animal care followed the European Union regulations (O.J. of E.C. L358/1 18 498 December 1986), and the European directive 2010/63/UE.

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
