## [Decision Letter]

Thank you for submitting your work entitled "A novel synaptic plasticity rule explains homeostasis of neuromuscular transmission" for consideration by *eLife*. Your article has been favorably evaluated by Eve Marder (Senior editor) and three reviewers, one of whom, Sacha Nelson, is a member of our Board of Reviewing Editors.

The reviewers have discussed the reviews with one another and the Reviewing Editor has drafted this decision to help you prepare a revised submission.

The following individuals involved in the review of your submission have agreed to reveal their identity: Benjamin Eaton and Alberto Cangiano (peer reviewers).

Summary:

The authors record from *Xenopus* neuromuscular junctions in culture and mouse ex vivo nerve-muscle preps and observe potentiation and depression which together implement a homeostatic learning rule for maintaining the gain of neuromuscular transmission close to 1. Both the potentiation and depression have been previously studied at the neuromuscular junction but here, by independently regulating pre and postsynaptic efficacy, the authors demonstrate a dual positive and negative feedback on presynaptic release mediated by calcium influx through nicotinic receptors and depolarization induced calcium release respectively.

Essential revisions:

1) The authors misstate the prior results of Wan and Poo who reported achieving LTP when the postsynaptic cell was either in voltage clamp or current clamp. The subsection “A stable synaptic gain” should entirely be rewritten since: 1) Wan and Poo (Science 1999) saw synaptic potentiation not only after presynaptic burst stimulation under myocyte Vclamp but also Iclamp, and 2) the statements of "…no apparent synaptic plasticity was visible. On the contrary…", are not clear. Also, the effects of nicotinic calcium signaling on presynaptic function presented here are seemingly in contrast to the previous observations (i.e. Dan and Poo, Science 1992) that found that increased nicotinic calcium produces synaptic depression and not LTP. The authors need to address these discrepancies.

2) The results presented on triggering LTP suggest that the sensitivity of the muscle to neurotransmitter is not what determines the increase in ACh release but rather the lack of the LTD mechanism induced by postsynaptic firing. This seems to be contrary to numerous studies from flies, mice, and humans supporting the idea that presynaptic compensation for changes in muscle sensitivity to neurotransmitter is one of the primary homeostatic mechanisms at the NMJ and driven by reduced AChR function or increasing K leak (see below). This point needs shoring up both with further discussion and with controls for the effects of curare incubation and/or changes in input conductance on ACh release in the absence of bursting. With respect to further discussion, the authors need to do a better job of explaining other differences between this data set and previous studies such as those of Paradis et al. 2001, Frank et al. 2006, Petersen et al. 1997; Cull-Candy et al. 1979 and 1980; Plomp et al. 1992 and Plomp and Molenaar 1996.

3) It is not clear whether the authors are suggesting that the positive and negative feedback mechanisms are dynamically balanced to achieve unity gain or gate each other to shut off plasticity once unity gain has been achieved. Given the fact that both forms of plasticity have previously been described and what is novel here is the interaction between the two it seems reasonable to insist on a better characterization of that interaction. This could be performed along one or both of the following dimensions:

A) Timing: Is the plasticity "gated" by SR calcium release or is there a temporal dependence to this mechanism? A tight temporal requirement would support that the gating mechanism is occurring within the muscle cell. Demonstrating a shared cellular localization for the convergence of these processes would strengthen the argument that LTD regulates/gates the LTP. Related to this point, is it surprising that the LTD is so stable in Figure 4 given that the ePSPs seem to all fail to reach threshold? Shouldn't this induce LTP?

B) Gradation: It is predicted that homeostatic regulation of PSP amplitudes would require a graded response to provide needed precision to the regulation (Davis, 2006). This should be demonstrated. Is either of the signals or their respective outputs graded? This could be revealed by investigating the freq dependence and/or the number of postsynaptic depolarizations required to generate LTD (Dan and Poo, Science 1992). Or is bulk unloading to the SR with thapsigargin sufficient to induce LTD? Overall, the authors need to strengthen their argument that this system is truly homeostatic. Similarly, is there a threshold level of nicotinic calcium required to induce LTP? Is there any freq dependence? Given previous observations from *Drosophila* and patients with myasthenia gravis, it might be expected that sub-blocking levels of curare would induce increased ACh release independent of bursting. Thus, an important control for the curare experiments is whether curare incubation in the absence of bursting or with chronic activity results in LTP.

One related concern is the safety factor for NMJ transmission – if homeostasis is ongoing wouldn't it tend to obliterate excess EPSP above that needed to evoke an AP?

Although it would be strongest to address both "axes" doing one or the other should be sufficient for publication (though the curare control experiment should be done in either case). This need not be carried out in both systems. Hence revision could involve a modest number of additional experiments which could be completed in two months or less.

---

## [Author Response]

Essential revisions:

*1) The authors misstate the prior results of Wan and Poo who reported achieving LTP when the postsynaptic cell was either in voltage clamp or current clamp. The subsection “A stable synaptic gain” should entirely be rewritten since: 1) Wan and Poo (Science 1999) saw synaptic potentiation not only after presynaptic burst stimulation under myocyte Vclamp but also Iclamp, and 2) the statements of "…no apparent synaptic plasticity was visible. On the contrary…", are not clear. Also, the effects of nicotinic calcium signaling on presynaptic function presented here are seemingly in contrast to the previous observations (i.e. Dan and Poo, Science 1992) that found that increased nicotinic calcium produces synaptic depression and not LTP. The authors need to address these discrepancies.*1A) *“Misstatement of the prior results of Wan & Poo”:*

In the pioneer study by Wan & Poo (Science, 1999) the induction of LTP was indeed shown both under postsynaptic current-clamp (Figure 1(i)) and voltage-clamp (Figure 1(ii-v)). Nonetheless, it was not clear to us whether the plasticity orientation rule (Figure 1 in Wan & Poo 1999), dependent on the initial synaptic conductance, was obtained under post-synaptic voltage- or current-clamp, since this information was not provided in the main text nor in the figure legend. Since the plasticity we found under current-clamp (Figure 6, blue dots) was different from that found by Wan & Poo 1999, we explicitly addressed the question whether this discrepancy was due to different clamping modes. We then performed the experiment under postsynaptic voltage-clamp, following the protocol of Figure 1(ii) in Wan & Poo. Figure 6 (black dots) shows the orientation rule obtained under voltage clamp, which perfectly matched the orientation rule found in Wan & Poo 1999 (Figure 1).

Author response image 1.Effect of burst stimulation of the motor neuron under postsynaptic voltage- and current-clamp.Black dots: same protocol as Wan & Poo 1999 (Figure 1 (ii)). Blue dots: same data as Figure 9 (black dots), expressed as a function of the initial averaged ePSCs amplitude.**DOI:**
http://dx.doi.org/10.7554/eLife.12190.016

In the single illustration of potentiation under postsynaptic current-clamp in Wan & Poo (Figure 1)), the synaptic efficacy during the conditioning burst was mainly sub-threshold (1 action-potential for the first event followed by 4 sub-threshold events composed the post-synaptic response to the bursting presynaptic stimulation). Figure 1 of our manuscript shows that given the correlation between the synaptic and the muscle input conductances (Figure 1), sub-threshold synaptic efficacy turned out to be rare and not representative of the initial conditions in our *Xenopus* cell culture. We then had to artificially reduce the ratio between the synaptic and the input conductances to obtain sub-threshold synaptic bursts that resulted systematically in potentiation, independently on the initial synaptic conductance (Figure 7, red dots).

All these observations brought us to conclude, maybe wrongly, that the orientation rule of the Figure 1 in Wan & Poo 1999 was established under postsynaptic voltage-clamp. However, since we think that this point is not crucial for our argumentation, we removed all mentions of voltage-clamp from the text.

Author response image 2.Degree of plasticity (relative change in ePSC amplitude) as a function of the initial ePSC.Potentiation obtained with a burst of subthreshold ePSPs (red dots) and depression obtained with direct triggering of the muscle APs (black dots). Same data than Figure 3 ('curare' bar) and Figure 4 ('direct AP' bar) expressed as a function of the initial averaged ePSCs amplitude.**DOI:**
http://dx.doi.org/10.7554/eLife.12190.017

The statements "…no apparent synaptic plasticity was visible. On the contrary.…", are not clear.

Under postsynaptic Iclamp, the bursting stimulation of the motor neuron did not induce drastic change in the synaptic gain in *Xenopus* (Figure 1) contrary to what was expected regarding the plasticity orientation rule obtained under post-synaptic Vclamp (Figure 6, black dots). This discrepancy was the object of the statements in the subsection “A stable synaptic gain”. Because of the correlation with the input conductance, most of synapses were efficient in eliciting the muscle spikes, and according to homeostatic rules, a synapse able to trigger the muscle spikes (i.e. close to the set point of the homeostasis) should not exhibit strong plasticity. This is further confirmed by the fact that chronic bursting stimulation of the motor neuron under post-synaptic Iclamp in non-treated synapses induced a low depression and extremely rarely a potentiation of the synaptic currents (Figure 6, blue dots).

We clarified the statement in the subsection “A stable synaptic gain”.

1B) *“Also the effects of nicotinic calcium signaling on presynaptic function presented here are seemingly in contrast to the previous observations (i.e. Dan and Poo, Science 1992) that found that increased nicotinic calcium produces synaptic depression and not LTP.”:*

Our study was more qualitative (nicotinic calcium versus DICR) than quantitative, and we estimate that our results do not contradict that “increased nicotinic calcium induces depression and not potentiation”. We think that our work is not in contrast with the idea that low calcium induces potentiation and high calcium depression, but points out how these two calcium regimes are physiologically achieved via different pathways.

Plasticity orientation rule under postsynaptic Vclamp supports the idea that “increased nicotinic calcium produces synaptic depression and not LTP”:

Even more than the data in Dan & Poo 1992, the orientation rule obtained under Vclamp may also suggest that “increased nicotinic calcium produces synaptic depression and not LTP”. Indeed, in Dan & Poo 1992, depression was obtained via ionophoretic ACh application. Since in the *Xenopus* cell culture the nicotinic receptors are not limited to the synaptic region, ionophoretic ACh application may mimic a competing synapse. This situation may be more related to synaptic competition and hetero-synaptic depression than to plasticity at mono-innervated muscle cell. Nicotinic calcium may have different effects when induced by another synapse. This idea is supported by the fact that Dan & Poo obtained depression with ionophoretic ACh alone and not when ACh was applied under postsynaptic Vclamp concomitantly with the pre-synaptic stimulation, while in this case nicotinic calcium build up should have been more important (Figure 1 of Dan & Poo 1992).

In contrast, under post-synaptic Vclamp at mono-innervated muscle cell, DICR is prevented (owing to the lack of depolarizations) and nicotinic calcium at the synapse is the only source of calcium. Since under Vclamp the driving force for Ca^2+^ is constant and the nicotinic calcium depends solely on the membrane permeability (i.e. the synaptic conductance), then nicotinic calcium increases linearly with the X-axis of Figure 6. Low nicotinic calcium-induced LTP and high nicotinic calcium-induced LTD may thus be an attractive explanation for the orientation of plasticity under post-synaptic Vclamp, and is compatible with the demonstration by Wan & Poo 1999 that potentiation depends on calcineurin (selectively activated by low calcium) while depression depends on CAMKII (selectively activated by high calcium).

The orientation rule in Wan & Poo 1999 (Figure 1), which matches our results under postsynaptic Vclamp (Figure 6, black dots), leaded to another interpretation: the initial synaptic currents amplitude may reflect the degree of maturity of the synapse, and immature synapses tend to potentiate while mature ones tend to depress. But we showed that 1) the synaptic current amplitude reflects in fact the adaptation to the post-synaptic input conductance and that the synapses are functionally equivalent (they trigger spikes), and 2) all the synapses are capable of both potentiation and depression. The potentiation shown in Figure 3 (“curare” bar) and the depression shown in Figure 4 (“direct AP” bar) were independent of the initial synaptic currents (Figure 7). Therefore, we believe that nicotinic calcium level-dependent plasticity is a more attractive explanation of the orientation of plasticity under post-synaptic Vclamp than a structural or functional capability of the synapses in plasticity.

Under postsynaptic Iclamp, the nicotinic calcium is low:

Are our data under post-synaptic Iclamp compatible with the view that low-calcium induces potentiation and high-calcium induces depression?

Under post-synaptic Iclamp, nicotinic calcium no longer depends solely on the synaptic conductance as in Vclamp but also on the variation of the membrane potential. We investigated the voltage dependency of the Ca^2+^ component of the nicotinic current in *Xenopus* myocytes (Figure 8) and we found the same Ca^2+^/V relationship as in Miledi et al. 1980 (J Physiol. 1980 Mar;300:197-212), i.e. the nicotinic Ca^2+^ exponentially decays with a potential constant of 30 mV. The nicotinic calcium build up is consequently strongly sensitive to the membrane potential. Thus for a given nicotinic conductance the nicotinic calcium is much lower under Iclamp than under Vclamp with a holding potential close to the resting potential.

In order to quantify the effect of the potential on the nicotinic calcium during a burst we did a conductance-based computer model based upon our characterization of the ionic currents of the *Xenopus* myocyte, including the voltage-dependency of the nicotinic calcium. For a given membrane surface, the increased permeability combined with the opposite voltage-dependency resulted in a bell-shape relationship between the nicotinic calcium and the synaptic conductance (Figure 8). As an illustration, simulations show that for a burst of 5 synaptic events at 20-30 Hz with a conductance producing a 1.5 nA synaptic current at a holding potential close to the resting potential, the use of Vclamp increases by 5 times the nicotinic calcium compared to Iclamp. In other words, under Iclamp the nicotinic calcium during this burst is equivalent of that provided under Vclamp by a much weaker synaptic conductance producing -300 pA of current. Therefore, the range of nicotinic calcium mobilized under Iclamp is globally in the range mobilized under Vclamp by low synaptic conductances which induce potentiation (Figure 6, black dots). Finally, the AP-induced DICR, known to be largest calcium signal in skeletal muscle cells, might be more susceptible than the nicotinic calcium to reach under IClamp the high level of calcium necessary to activate CAMKII and to induce depression.

The corollary could be that under Vclamp, the nicotinic calcium artificially reaches the threshold inducing depression because of the maximum driving force maintained by the holding potential.

Author response image 3.Voltage-dependency of the nicotinic calcium.(**A**) Nicotinic I/V relationship with iohophoretic ACh applications in media for which the Ca^2+^ ion only carries the nicotinic current. Na^+^ and Mg^2+^ were removed, K^+^ was adjusted for each holding potential in order to remain at the equilibrium, and Lysine hydrochloride was inversely adjusted to KCl in order to hold both the osmotic pressure and the ionophoretic ACh application. Dashed line shows the rectification predicted with the GHK free diffusion model. (B) Example for a given cell surface of simulation of the nicotinic calcium in V- and I-clamp. The model comprised a synapse, a linear passive K^+^ leak, an inward rectifying K^+^ leak, two voltage-gated K^+^ conductances and a voltage-gated Na^+^ conductance.**DOI:**
http://dx.doi.org/10.7554/eLife.12190.018

In conclusion, our results under Iclamp seem to not be in contrast with the fact that “increased nicotinic calcium induced depression and not potentiation”. We thought interesting to discuss here the discrepancies between Iclamp and Vclamp, but the issue of Vclamp being not crucial to the paper, we chose to avoid a lengthy discussion on this point and we suppressed its mention from the text. We added to the Discussion of the revised version of the manuscript that the low level of nicotinic calcium under Iclamp and the high level of calcium reached by the DICR signal are consistent with the demonstration by Wan & Poo of the roles of calcineurin and CAMKII in plasticity.

*2) The results presented on triggering LTP suggest that the sensitivity of the muscle to neurotransmitter is not what determines the increase in ACh release but rather the lack of the LTD mechanism induced by postsynaptic firing. This seems to be contrary to numerous studies from flies, mice, and humans supporting the idea that presynaptic compensation for changes in muscle sensitivity to neurotransmitter is one of the primary homeostatic mechanisms at the NMJ and driven by reduced AChR function or increasing K leak (see below). This point needs shoring up both with further discussion and with controls for the effects of curare incubation and/or changes in input conductance on ACh release in the absence of bursting. With respect to further discussion, the authors need to do a better job of explaining other differences between this data set and previous studies such as those of Paradis et al. 2001, Frank et al. 2006, Petersen et al. 1997; Cull-Candy et al. 1979 and 1980; Plomp et al. 1992 and Plomp and Molenaar 1996.*

2A) Controls in absence of bursting:

The main concern raised in this point seems to be that we used evoked neurotransmitter release (in the form of trains of presynaptic stimulations called “bursts”) to induce potentiation, while in the *Drosophila* larva evoked release is not necessary to a rapid compensatory potentiation in response to the reduction of the post-synaptic sensitivity (Frank et al. 2006). If evoked release is not necessary for potentiation in *Drosophila*, synaptic activity is nonetheless required, since the authors propose that spontaneous release remaining in absence of motor neuron activity is responsible for this potentiation.

Following the reviewer suggestions, we performed two control experiments:

1) In our potentiation induction, it has to be noted that curare was transiently (1 min or less) applied to reduce the synaptic efficacy during the conditioning burst. In order to test that evoked bursts, and not spontaneous release, was responsible for potentiation, we transiently applied curare during 2 min in absence of pre-synaptic stimulation, and we did not obtain potentiation (relative change in ePSC of 0.89 ± 0.05, n=3). We have added these data to Figure 3 (“Curare No burst” bar).

2) We did another control with longer lasting incubation of curare. Given the large range of initial synaptic conductances (Gsyn) in the cell culture but the strong correlation (Figure 1) between this conductance with the input conductance (Gin), we compared the Gsyn/Gin ratio in non-treated synapses with synapses incubated in curare after 30-60 min of exposure. The averaged Gsyn/Gin ratio in curare (Gsyn/Gin = 1.35 ± 0.32, n = 8) was significantly lower than the averaged Gsyn/Gin ratio in non-treated synapses (Gsyn/Gin = 3.78 ± 0.49, n = 10) with a significance level of 0.001. Since motor neurons do not spontaneously fire in the *Xenopus* cell culture, this result shows that within 60 min of curare incubation the spontaneous ACh release was unable to sufficiently promote the evoked release in order to restore a normal evoked Gsyn/Gin ratio.

2B) The frequency of spontaneous synaptic activity may explain some differences between *Drosophila* and vertebrates:

The 60 min control incubation of the above experiment does not presume the outcome of a longer curare exposure that could result in a full recovery of the synapse efficacy. In the *Xenopus* cell culture, motor neurons are “isolated” (not connected by afferences) and do not spontaneously fire. Despite the absence of motor neurons spikes, they connect the muscle cells and the mechanisms responsible for the evoked release establish during the 12-24h before the recordings. The establishment of a functional evoked release in absence of neuronal spikes, and moreover dependent on the muscle input conductance (Figure 1), presumably implicates the spontaneous release. The difference between *Drosophila* and *Xenopus* in the speed of compensatory potentiation induced by spontaneous activity may simply be related to the strong difference between these systems in the frequency of the spontaneous activity. In Paradis et al. 2001, Frank et al. 2006 or Petersen et al. 1997, it seems that the frequency of spontaneous release in *Drosophila* larva is around 10-20 Hz, while it is 100 time less on average in our *Xenopus* cell culture.

2C) Evoked activity should also mobilize homeostatic mechanisms:

If spontaneous synaptic events are able to mobilize some synaptic plasticity mechanisms, it seems reasonable to envisage that evoked release does too. Moreover, homeostasis of the neuromuscular transmission adapts the evoked neurotransmitter release to the post-synaptic input conductance in order to ensure the reliability of the motor command transmission. Therefore, it seems useful for the homeostatic mechanisms that evoked release participates to the evaluation of its own efficacy and that this evaluation does not depend solely on the spontaneous release. The large literature produced by the Poo’s group, and the present work, suggest that in *Xenopus* cell culture bursts of evoked activity are able to trigger the plastic properties of the synapse more rapidly than spontaneous activity.

In *Drosophila*, Petersen et al. or Paradis et al. observed the outcome of the homeostasis but it does not exclude that evoked neurotransmitter release had participated to the homeostasis before its observation. Even if Frank et al. show that 10-20 Hz spontaneous activity is able to trigger potentiation in absence of evoked activity, this does not prove that evoked activity with the frequency of a normal motor command cannot do the same. A possible stabilizing role of synaptic depressing processes mobilized by evoked activity remains to be investigated but is not yet excluded in *Drosophila*.

2D) Evoked burst at 30 Hz is natural:

In vertebrate, evoked activity is responsible for the muscular tonus and voluntary motions, and represents the prominent part of synaptic activity. In vertebrates, the frequency range of a motor command is 10-60 Hz, and muscle activity being made of temporary motions, the motor command is made of bursts. Thus, the conditioning bursts we used in our experimental protocols closely mimic the typical firing rate of individual motor units, as recorded in fast and slow muscles in the conscious rat during unrestrained walking (Gorassini et al., 2000 Journal of Neurophysiology 83 (4): 2002–11). In addition, bursting activity with frequency of action potentials in the range of 10-60 Hz has also been observed in the developing spinal cord (L. T. Landmesser and M. J. O’Donovan, 1984, J. Physiol. London347, 189; M. J. O’Donovan et al., 1998, Ann. N.Y. Acad. Sci.860, 130).

2E) *The results presented on triggering LTP suggest that the sensitivity of the muscle to neurotransmitter is not what determines the increase in ACh release but rather the lack of the LTD mechanism induced by postsynaptic firing.*

This question is related to the point 3 of the essential revisions, concerning the type of interaction between the potentiating and the depressing synaptic mechanisms. We used the extreme cases of subthreshold synaptic events and ryanodine treated preparations to prevent DICR and to emphasize the potentiating role of the nicotinic calcium. But as discussed in the answer to point 3, and as shown in the new Results section of the revised version (Figure 5), potentiation also occurs in a range of synaptic efficacy mixing nicotinic calcium with DICR signals. We show in this new section that potentiation is more related to how far from an “attractor Gsyn/Gin ratio” is the initial Gsyn/Gin than it is due to “the lack of depressing mechanism”.

One goal of our work was to show that suprathreshold synaptic activity mobilizes both potentiating and depressing mechanisms, and that homeostasis might be due to a balance between these antagonist mechanisms. Even if synaptic strength stabilization at a set point results from such balance, the “postsynaptic sensitivity to the neurotransmitter” – here assimilated to the ratio Gsyn/Gin– remains “one of the primary variables determining presynaptic compensation”.

2F) *Differences between this data set and previous studies such as those of Paradis et al. 2001, Frank et al. 2006, Petersen et al. 1997; Cull-Candy et al. 1979 and 1980; Plomp et al. 1992 and Plomp and Molenaar 1996.*:

Beyond the difference between vertebrates and *Drosophila* concerning the frequency of spontaneous synaptic activity, there are other parameters that make a direct comparison of these systems difficult. In particular, given the lack of muscle action potentials and DICR signal in the *Drosophila* larva, the mechanism we propose here cannot be directly transposed from vertebrate to *Drosophila*. However, it could well be that in *Drosophila* the evoked release mobilize a putative negative side of the homeostasis through actors other than DICR. The literature on *Drosophila* focuses mainly on the presynaptic expression of the homeostatic potentiation, and not on the postsynaptic mechanisms of evaluation of synaptic efficacy leading to the homeostasis. However, despite the absence of muscle spikes in *Drosophila*, evoked postsynaptic potentials should activate low-voltage-activated Ca^2+^ channels and the calcium-induced calcium release responsible for muscle contraction. It is tempting to speculate on the possible role of these calcium signals in the stabilization of the evoked release in *Drosophila*, and this remains to be investigated. We added to the Discussion the comparison between *Drosophila* and our data, including the differences in the role of the spontaneous release.

In the study by Cull-Candy et al. 1979 and 1980, the authors compared the neurotransmitter release in Myasthenia Gravis (MG) and control patients. They found that the neurotransmitter release was not responsible for the weakness characteristic of the MG disease (Cull-Candy et al. 1979), and that in contrast a five-fold increase in the quantal content of the evoked release partially compensated the reduced number of postsynaptic receptors (Cull-Candy et al. 1980). Therefore, this work is a post-observation of the outcome of a compensatory process which had occurred into the organism, and does not predict what kind of synaptic activity between spontaneous release and evoked motor command was responsible in the triggering of such compensation. Despite their weakness, MG patients’ movements are indeed due, as mentioned above, to bursts of evoked release in a frequency range containing the 30Hz we used in our experiments. Therefore, the increase in the quantal content we obtained in response to reduced Gsy/Gin and the mechanism we propose seem not in contrast with these observations.

A similar argument can be applied for the study by Plomp et al. 1992 and Plomp & Molenaar 1996. In this work, the author mimicked the MG syndrome in rat with chronic injections of α-bungarotoxin, and observed that an increase in the evoked release compensate for the reduced number of functional postsynaptic receptors, an observation that is not in contrast with our work. The time course of the compensation cannot be compared with our work since bungarotoxin was injected chronically every 48h. However, a difference with our work is the apparent long delay (3h minimum) before the observation of the compensation. This apparent delay might be linked to the mode of bungarotoxin administration, i.e. subcutaneous injection, which renders the time course of the appearance of the drug’s effect at the diaphragm neuromuscular junctions difficult to predict. We introduced these references, and these elements of discussion, to the Discussion section of the revised manuscript.

*3) It is not clear whether the authors are suggesting that the positive and negative feedback mechanisms are dynamically balanced to achieve unity gain or gate each other to shut off plasticity once unity gain has been achieved. Given the fact that both forms of plasticity have previously been described and what is novel here is the interaction between the two it seems reasonable to insist on a better characterization of that interaction. This could be performed along one or both of the following dimensions: A) Timing: Is the plasticity "gated" by SR calcium release or is there a temporal dependence to this mechanism? A tight temporal requirement would support that the gating mechanism is occurring within the muscle cell. Demonstrating a shared cellular localization for the convergence of these processes would strengthen the argument that LTD regulates/gates the LTP. Related to this point, is it surprising that the LTD is so stable in Figure 4 given that the ePSPs seem to all fail to reach threshold? Shouldn't this induce LTP? B) Gradation: It is predicted that homeostatic regulation of PSP amplitudes would require a graded response to provide needed precision to the regulation (Davis, 2006). This should be demonstrated. Is either of the signals or their respective outputs graded? This could be revealed by investigating the freq dependence and/or the number of postsynaptic depolarizations required to generate LTD (Dan and Poo, Science 1992). Or is bulk unloading to the SR with thapsigargin sufficient to induce LTD? Overall, the authors need to strengthen their argument that this system is truly homeostatic. Similarly, is there a threshold level of nicotinic calcium required to induce LTP? Is there any freq dependence? Given previous observations from Drosophila and patients with myasthenia gravis, it might be expected that sub-blocking levels of curare would induce increased ACh release independent of bursting. Thus, an important control for the curare experiments is whether curare incubation in the absence of bursting or with chronic activity results in LTP.*

*One related concern is the safety factor for NMJ transmission – if homeostasis is ongoing wouldn't it tend to obliterate excess EPSP above that needed to evoke an AP?* 3A) *“Positive and negative feedback mechanisms are dynamically balanced to achieve unity gain or gate each other to shut off plasticity once unity gain has been achieved?”*

Some elements for answering this question may already be found in the data of the paper, and they support the dynamical balance hypothesis. It should be noted that despite an apparent safety factor (slope in Figure 1 ≥ 3, Figure 9), the synaptic gain in the *Xenopus* cell culture is not strictly 1 as it is in the mouse neuromuscular junction, but rather close to 0.7-0.8 (Figure 1). This results from the variability of neurotransmitter release in this model (Figure 1). Evoked post-synaptic activity is thus made of 70-80% of action potentials (some triggered by release largely exceeding the threshold) and 20-30% of sub-threshold ePSPs. The bursts composing the chronic activity in Figure 1 were made of these proportions of sub- and supra-threshold events. We reasoned that in the case of a “gating mechanism”, supra-threshold events would have been neutral in terms of plasticity (“shut off plasticity”), and thus the remaining 20% of sub-threshold events would have potentiated the synapse up to gain 1. Instead of that, the synaptic gain slightly decreased (Figure 1) due to a slight depression of the averaged synaptic currents (0.2, subsection “A stable synaptic gain “of the initial manuscript). The absence of potentiation despite the presence of sub-threshold events is not in favor of “gating” hypothesis, and the plasticity outcome – i.e. depression – excludes it.

Gradation of depression:

To examine these issues in more details, we performed new experiments addressing the plasticity response to chronic bursting activity in non-treated *Xenopus* synapses (same protocol than in Figure 1). The small depression observed after chronic activity was not homogenous among tested synapses (Figure 9, black dots). Interestingly, the degree of depression of a given synapse seems dependent on how far its initial Gsyn/Gin ratio was from the averaged ratio of the population measured after chronic activity (Figure 9, inset, dotted line). Consequently, standard deviation from the mean diminished after chronic activity. We interpret this synaptic behavior as a convergence towards the set point of homeostasis, induced by the evoked bursting activity.

Author response image 4.Homeostatic control of the synaptic efficacy.(**A**) In *Xenopus*, ratios between averaged synaptic conductance ('Gsyn', calculated from 30–40 ePSCs) and muscle cell input conductance ('Gin') before and after 20-30 min of chronic burst stimulation of the motor neuron (burst of 20–60 events at a 20–30 Hz frequency, every 30–40 s) under postsynaptic current-clamp in non-treated (black dots) and low curare-treated (red dots) synapses. Green dots represent the Gsyn/Gin ratio before and after 1–3 bursts of 5 presynaptic stimulations at 30 Hz in ryanodine loaded muscle cells (same data than in Figure 3, ryanodine bar). Inset show the mean ± standard deviation of the Gsyn/Gin ratios in the three conditions. The dotted lines show the mean Gsyn/Gin ratio after chronic activity in non-treated synapses. (**B**) Degree of plasticity shown in A expressed as a function of the difference between the initial individual Gsyn/Gin ratio and the mean ratio after chronic burst activity ('Distance to the set point'), in non-treated (black dots) and curare-treated (red dots) synapses. The solid line shows the theoretical relationship between plasticity and the distance to a set point of 2.36, calculated as the mean Gsyn/Gin ratio after chronic activity in non-treated synapses.**DOI:**
http://dx.doi.org/10.7554/eLife.12190.019

Gradation of potentiation:

In order to test whether synapses also converge after that the initial Gsyn/Gin ratio was decreased below the set point, we applied the same chronic activity in the continuous presence of low doses of curare (0.1–2 µM), as proposed by the reviewer (Figure 9, red dots). In presence of curare, chronic burst stimulation induced convergence of synapses towards the same set point, with a potentiation magnitude also depending on how far the initial Gsyn/Gin ratio was from the averaged ratio after chronic activity.

“A truly homeostatic system”:

In a perfect homeostatic process, with an attractor set point, the degree of plasticity (relative change in the Gsyn/Gin ratio) can be expressed as a function of the distance to the set point (difference between the initial Gsyn/Gin ratio and the set point):degreeofplasticity=1−11+setpointdistance

Figure 9 shows this theoretical relationship (line) with a set point of 2.36 (the averaged Gsyn/Gin ratio obtained after chronic activity). In order to confront our data with a perfect theoretical homeostasis, we added to Figure 9 the degrees of plasticity in non-treated (black dots) and curare treated (red dots) synapses shown in A, expressed as a function of the difference between the initial Gsyn/Gin ratio and the 2.36 set point.

In conclusion, gradation in both potentiation and depression could be obtained in curare-treated and non-treated synapses respectively, suggesting that this system “is truly homeostatic” and that potentiating and depressing mechanisms balance rather than gate each other.

Why is the strength of *Xenopus* synapses, in initial conditions (12-24 hours after cultivation start), correlated to the postsynaptic input conductance above the target set point reached after evoked activity? As discussed in the answer to point 2 of the questions of the reviewers, *Xenopus* motoneurons in culture do not spontaneously fire and the mechanisms responsible for the evoked ACh release take place under the influence of the spontaneous release, which has lower amplitudes (a minority reaches the threshold). The potentiation/depression balance mobilized by spontaneous release is in favor of potentiation, and this may explain that the evoked release is found in initial conditions above the normal set point.

The effects of ryanodine in *Xenopus* and mouse in regards to the set point of homeostasis:

In order to place the potentiation, we obtained with ryanodine treatment in regards to the homeostatic law described above, we normalized by Gin the synaptic conductances found in ryanodine-loaded *Xenopus* muscle cells in Figure 3 (“Ryanodine” bar), and we added these data to Figure 9 (green dots).

In adult mouse neuromuscular synapses, an apparent Gsyn/Gin ratio can be estimated from the ePSPs in the following manner. If we assume a linear passive leak, the Gsyn/Gin ratio can be expressed as:

GsynGin=−80−VpVp with Vp the membrane potential reached by the ePSP and 0 and -80 mV the reversal potentials of the synaptic and the leak currents respectively. Figure 10 shows the apparent Gsyn/Gin ratio in non-stimulated and in burst-stimulated preparations, both for non-treated and ryanodine-treated preparations. Contrary to *Xenopus* where the membrane of the muscle cells is isopotential, these ratios in mouse are only apparent since the recorded ePSP amplitude depends on the distance between the synapse and the recording site. This distance effect is limited in FDB muscle because the cells are “only” 300 µm long, but nonetheless the apparent Gsyn/Gin ratio is presumably lower than the actual ratio at the synapse location. For this technical reason, the distance effect might also participate to the variability in apparent ePSPs amplitudes.

Both in mouse and *Xenopus*, the ryanodine-treated synapses did not converge towards a set point. The averaged Gsyn/Gin after burst activity was higher than the set point, and the standard deviation from the mean was increased. These data suggest that the ryanodine receptors-dependent calcium signal participates to a dynamical balance stabilizing the synaptic efficacy at a set point.

Author response image 5.Plasticity in mouse.(**A**) Apparent averaged Gsyn/Gin ratios calculated in mouse FDB muscles from the data of Figure 3 with equation X, in non-stimulated and non-treated synapses (no burst, black dot), in burst-stimulated and non-treated synapses (after chronic bursts, black dot), in non-stimulated and ryanodine-treated synapses (no burst, green dot) and in burst-stimulated and ryanodine-treated synapses (after chronic bursts, green dot). Dots represent the mean ± Standard Deviation. (**B**) Relative change of contraction force during 2s-30Hz bursts of nerve stimulations in mouse soleus muscles (n=4), before and during exposure of a low dose (0.1 µM) of curare.**DOI:**
http://dx.doi.org/10.7554/eLife.12190.020

The goal of this work was to show that the physiological range of nicotinic calcium plays a potentiating role on the synaptic strength, that the AP-induced calcium release from the reticulum plays a depressing role, and that this depressing effect participates to stabilization of the neurotransmitter release in a supra-threshold range. However, important questions such as the precise roles of the two calcium signals in the balance, and the location(s) where the push-pull mechanism(s) acts require more experiments.

We have included the new data in the Results of the revised version of our manuscript under the title “Plasticity orientation rule is homeostatic”. A newly added figure contains the data presented above as well as a low curare experiment (0.1 µM) done on the mouse nerve-muscle preparation in order to show that the mouse neuromuscular synapse is also capable of compensatory potentiation under curare.

In this last data set, we took advantage of the fact that the muscle contraction force during a presynaptic burst integrates the synaptic efficacy over all the synapses and all the synaptic events. The Figure 10 shows that burst stimulation of the nerve triggers a compensatory potentiation that results in the recovery of a normal contraction force.

3B) *Timing between nicotinic calcium and DICR:*

The timing approach is another interesting suggestion, but we think that this approach is more related to hetero-synaptic depression and synaptic selection than to homeostasis at a mono-innervated muscle cell. In order to show that the calcium release from reticulum through the ryanodine receptors induce synaptic depression we have directly triggered the muscle AP in absence of pre-synaptic stimulation. However, in the natural circumstances this situation occurs only in multi-innervated muscle cells, during development. In our work we did not investigate the possible role of the AP-associated calcium signal in heterosynaptic depression, a hypothesis which remains to be tested. It is tempting to hypothesize that if the global DICR signal (activated across the whole fiber) is associated with the release by all the muscle surface of a negative retrograde feedback factor, it could participate to the depression of the less active synapses in the case of a competition. Interestingly, *Drosophila* larva muscles, that do not possess the global DICR signal of the vertebrates, are multi-innervated.

3C) *Is it surprising that the LTD is so stable in Figure 4 given that the ePSPs seem to all fail to reach threshold? Shouldn't this induce LTP?*

Actually, it does, but slowly. Figure 11 shows in a muscle, for which the nerve was relatively regularly stimulated at low rate, the slow increase in ePSPs amplitudes (r=0.23, p=0.07).

Author response image 6.In mouse, slow potentiation after depression.ePSPs peak in individual FDB cells after depression induced by bursts of external stimulations in a Ca^2+^ free medium.**DOI:**
http://dx.doi.org/10.7554/eLife.12190.021

As we discussed in our answer to point 2 of the reviewers, we attribute the apparent absence of potentiation (or slow potentiation) in Figure 4 to the low rate of stimulation used to measure the ePSPs. In the course of the 60 min test protocol, in Figure 4 and Figure 11, each dot after the external stimulation correspond to ePSP peak in response to a discrete “test” stimulation of the nerve. ePSPs are sequentially recorded in different muscle fibers, resulting in a low rate stimulation of the nerve. The Figure 4—figure supplement 4 of the initial submission (Figure 4—figure supplement 3 in the revised version) was dedicated to show that rapid LTP is indeed inducible after depression as expected by the reviewer, if higher frequency of nerve stimulation is used (arrows, bursts of nerve stimulations at 20-30Hz). Similarly, in *Xenopus*, potentiation is obtained by motoneuron stimulation (1-3 bursts of 5 events at 30Hz) after depression induced by direct muscle AP (1-3 bursts of currents steps into the muscle cells at 30Hz) (not shown). For clarity, we removed the Figure 3—figure supplement 3 where we did not wait enough in *Xenopus* to restore the normal synaptic gain after depression.

3D) *Dependence on Frequency and number of postsynaptic events*

We did not attempt to perform such studies for the revised paper. Instead we have focused on the new experiments described above, that show the graded behavior of plasticity for both depression and potentiation.

3E) *Given previous observations from Drosophila and patients with myasthenia gravis, it might be expected that sub-blocking levels of curare would induce increased ACh release independent of bursting.*

This point is directly related to the point 2 of the essential revisions. Please see our detailed answer to point 2.

3F) *Is there a threshold level of nicotinic calcium required to induce LTP?*

The apparent threshold number and frequency of stimulations necessary to induce potentiation in Wan & Poo 1999 (Figure 3) could suggest that there is a threshold of nicotinic calcium build-up required to induce rapid potentiation. Related to this point, we detailed in the point 1 of the essential revisions that the nicotinic calcium is strongly dependent on the membrane potential. For plasticity induced under post-synaptic Vclamp, in contrast with a -80 mV holding potential, the temporary use of a -50 mV holding during the conditioning burst reveals a threshold synaptic conductance to induce a rapid potentiation (Figure 12), suggesting that a threshold level of nicotinic calcium is required.

Author response image 7.Dependency of plasticity under postsynaptic Vclamp on the holding value during the conditioning burst.Same protocol than Wan & Poo 1999 (Figure 1 (ii)) with different holding potentials during the conditioning burst.**DOI:**
http://dx.doi.org/10.7554/eLife.12190.022

3G) *One related concern is the safety factor for NMJ transmission – if homeostasis is ongoing wouldn't it tend to obliterate excess EPSP above that needed to evoke an AP?*

Yes, in the case of a gating mechanism. But we argued in the text above that a dynamical balance is more likely than a gating. In the case of a dynamical balance, the mean level of synaptic conductance reaching the equilibrium is difficult to predict when we don’t know the location(s) and the nature(s) of the push-pull mechanism(s). Equilibrium is not necessarily reached at the spike threshold.